# High-resolution structures of the actomyosin-V complex in three nucleotide states provide insights into the force generation mechanism

**Sabrina Pospich[1], H Lee Sweeney[2], Anne Houdusse[3], Stefan Raunser[1]\***

[1]Department of Structural Biochemistry, Max Planck Institute of Molecular Physiology, Dortmund, Germany; [2]Department of Pharmacology and Therapeutics and the Myology Institute, University of Florida, Gainesville, United States; [3]Structural Motility, Institut Curie, Centre National de la Recherche Scientifique, Paris, France

**Abstract** The molecular motor myosin undergoes a series of major structural transitions during its force-producing motor cycle. The underlying mechanism and its coupling to ATP hydrolysis and actin binding are only partially understood, mostly due to sparse structural data on actin-bound states of myosin. Here, we report 26 high-resolution cryo-EM structures of the actomyosin-V complex in the strong-ADP, rigor, and a previously unseen post-rigor transition state that binds the ATP analog AppNHp. The structures reveal a high flexibility of myosin in each state and provide valuable insights into the structural transitions of myosin-V upon ADP release and binding of AppNHp, as well as the actomyosin interface. In addition, they show how myosin is able to specifically alter the structure of F-actin.

**\*For correspondence:**
stefan.raunser@mpi-dortmund.mpg.de

**Competing interest:** The authors declare that no competing interests exist.

## Editor's evaluation

This work obtains an atomic-level understanding of mechanochemical coupling and the structural elements that lead to mechanical and chemical diversity of the cytoskeletal motor, myosin-V.

## Introduction

The molecular motor myosin is well known for its central role in muscle contraction (*Hanson and Huxley, 1953*; *Szent-Györgyi, 2004*). By using the actin cytoskeleton as tracks, myosin also powers cellular cargo transport processes and can serve as a molecular anchor and force sensor (*Hartman et al., 2011*; *Woolner and Bement, 2009*). Due to its versatility, myosin is key to numerous essential cellular processes including cytokinesis, transcription, signal transduction, cell migration and adhesion, and endo- and exocytosis (*Coluccio, 2020*; *Krendel and Mooseker, 2005*). While this variety in functions is well reflected by the diversity of the myosin superfamily (*Sellers, 2000*), the ATP-dependent force generation mechanism as well as the architecture of the motor domain is shared by all myosins (*Cope et al., 1996*).

The myosin motor domain consists of four subdomains: the actin-binding upper and lower 50 kDa (U50 and L50) domains, which are separated by the central actin-binding cleft, the N-terminal domain, and the converter domain, containing the long α-helical extension known as the lever arm (*Rayment et al., 1993b*). The active site of myosin is located at the interface of the U50 domain and the N-terminal domain and is allosterically coupled to both the actin-binding interface and the lever arm (*Sweeney and Houdusse, 2010*). This coupling ultimately enables the amplification of small

rearrangements at the active site to large, force-producing conformational changes of the lever arm (*Holmes, 1997*; *Rayment et al., 1993a*).

The ATP-driven mechanism of myosin force generation relies on several major structural transitions and is described in the myosin motor cycle (*Huxley, 1958*; *Lymn and Taylor, 1971*). Initially, hydrolysis of ATP places myosin in a conformation known as the pre-powerstroke (PPS) state. The mechano-chemical energy stored in this conformation is released by binding to filamentous actin (F-actin), which serves as an activator and initiates a cascade of allosteric structural changes (*Rosenfeld and Sweeney, 2004*; *Walker et al., 2000*). These changes eventually result in phosphate release—potentially via a phosphate release ($P_iR$) state (*Llinas et al., 2015*)—and the major, force-producing lever arm swing known as the powerstroke. Subsequent release of ADP from myosin in a state that binds both F-actin and ADP strongly (strong-ADP state) gives rise to a second, smaller lever arm swing, leaving myosin strongly bound to F-actin in the rigor state (*Whittaker et al., 1995*; *Mentes et al., 2018*). Binding of ATP to the now unoccupied active site causes a transition to the post-rigor state and eventual detachment from F-actin (*Kühner and Fischer, 2011*). Finally, ATP hydrolysis triggers the repriming of the lever arm through the so-called recovery stroke, thus completing the myosin motor cycle.

Decades of biochemical studies have brought great insights into the diversity and kinetics of the myosin superfamily (*Coluccio, 2020*; *Geeves et al., 2005*). However, detailed structural information is ultimately required to understand the mechanism of force generation. Over the years, X-ray crystallography has revealed the structures of various myosins in the post-rigor state (*Rayment et al., 1993b*), the PPS state (*Smith and Rayment, 1996*), the rigor-like state (*Coureux et al., 2003*), a putative $P_iR$ state (*Llinas et al., 2015*), as well as the intermediate recovery stroke state (*Blanc et al., 2018*; for a recent review of all available crystal structures, see *Sweeney et al., 2020*). Due to the reluctance of F-actin to crystallize, actin-bound states of myosin are not accessible by X-ray crystallography. Instead, cryo electron microscopy (cryo-EM) has proven to be an optimal tool to study filamentous proteins (*Pospich and Raunser, 2018*) such as the actomyosin complex (*Behrmann et al., 2012*; *von der Ecken et al., 2016*). To date, the structure of the actomyosin rigor complex has been determined for a variety of myosins (*Banerjee et al., 2017*; *Behrmann et al., 2012*; *Doran et al., 2020*; *Fujii and Namba, 2017*; *Gong et al., 2021*; *Gurel et al., 2017*; *Mentes et al., 2018*; *Risi et al., 2021*; *Robert-Paganin et al., 2021*; *Vahokoski et al., 2020*; *von der Ecken et al., 2016*). States other than the nucleotide-free rigor state have proven more difficult to study, mainly due to lower binding affinities and short lifetimes. In fact, the only other state solved to date is the strong-ADP state; and only two (myosin-IB, myosin-XV) (*Gong et al., 2021*; *Mentes et al., 2018*) of four independent studies (myosin-Va, myosin-VI) (*Gurel et al., 2017*; *Wulf et al., 2016*) have achieved high resolution (<4 Å). However, the actin-bound states of myosin, in particular weakly bound transition states for which no structure is yet available, are precisely those that are urgently needed to understand important properties of the myosin motor cycle, such as binding to and detachment from F-actin (recently reviewed in *Schröder, 2020*). In addition, high-resolution structures of other myosins in the rigor and especially strong-ADP state are required to identify conserved and specific features within the myosin superfamily. Finally, structures of all key states of the motor cycle need to be determined for a single myosin to allow the assembly of a reliable structural model since the structures of different myosins vary considerably within the same state (*Merino et al., 2020*).

Some myosins, including myosin-IB and the high-duty ratio myosins V and VI, have comparatively high binding affinities for F-actin and long lifetimes of actin-bound states (*De La Cruz and Ostap, 2004*; *De La Cruz et al., 2001*; *De La Cruz et al., 1999*; *Laakso et al., 2008*). Therefore, they are best suited to structurally study actin-bound states other than the rigor. Today, class V and VI myosins are probably the best-characterized unconventional myosins, both structurally and biochemically (*Coluccio, 2020*). Cryo-EM studies of actomyosin-V have further reported structures of the strong-ADP and rigor state (*Wulf et al., 2016*), as well as a potential PPS transition state (*Volkmann et al., 2005*). However, due to the limited resolution of these structures, atomic details could not be modeled and the structural transition of actin-bound myosin-V during its motor cycle has consequently remained elusive. Interestingly, myosin-V was also shown to be sensitive to the nucleotide state of phalloidin (PHD)-stabilized F-actin, preferring young ATP/ADP-$P_i$-bound F-actin over aged (post-$P_i$ release) ADP-bound F-actin (*Zimmermann et al., 2015*). The structural basis and implications of this preference have not yet been uncovered.

**Table 1.** Data collection statistics of F-actin and actomyosin data sets.

Aged PHD-stabilized F-actin (F-actin-PHD) was decorated with myosin-V in the rigor (no nucleotide), strong-ADP (bound to Mg$^{2+}$-ADP) and post-rigor transition (PRT) state (bound to Mg$^{2+}$-AppNHp). Young JASP-stabilized F-actin (F-actin-JASP) was imaged in absence and presence of myosin-V in the rigor state. Refinement and model building statistics can be found in *Table 2*, *Table 3*, *Table 4* and *Table 6*. See *Figure 1—figure supplement 1* for an overview of the processing pipeline.

| | Aged F-actin-PHD | | | | | Young F-actin-JASP | |
| --- | --- | --- | --- | --- | --- | --- | --- |
| | **ADP** | **Rigor** | **AppNHp 4°C** | **AppNHp 25°C** | **AppNHp\*** | **Actin only** | **Rigor** |
| **Microscopy** | | | | | | | |
| Microscope | Titan Krios – Cs 2.7 mm | | Titan Krios – Cs-corrected | | | | |
| Voltage (kV) | | | 300 | | | | |
| Camera | | | K2 – super resolution | | | | |
| Energy filter slit width (eV) | | | 20 | | | | |
| Pixel size (Å) | 1.06 | | | 1.10 | | | |
| Frames per movie | | | 40 | | | | |
| Exposure time (s) | | | 15 | | | | |
| Total electron dose (e/Å$^2$) | 79 | 82 | 81 | 81 | 81 | 80 | 80 |
| Final electron dose (e/Å$^2$) | | | Dose weighted | | | Polished particles | |
| Defocus range (μm) | 0.3–3.2 | 0.5–3.0 | 0.3–3.0 | 0.3–3.0 | 0.3–3.0 | 0.3–2.9 | 0.3–3.0 |
| Number of images† | 4571 (5908) | 2304 (3623) | 5858 (7121) | 6617 (7023) | 12,475 | 936 (1064) | 2970 (3336) |

\*Combined from two data sets (4°C and 25°C).
†In parenthesis is the initial number of images.

Here, we present high-resolution cryo-EM structures of the actomyosin-V complex in three nucleotide states. Specifically, we have solved the structure of myosin-V in the strong-ADP state (ADP), the rigor state (nucleotide free), and a previously unseen post-rigor transition (PRT) state, which has the non-hydrolyzable ATP analog AppNHp bound to its active site. To investigate the structural effect the nucleotide state of F-actin has on myosin-V, we have also determined the structure of the rigor complex starting from young ADP-P$_i$-bound F-actin, rather than from aged ADP-bound F-actin. In addition to these structures and their implications, we report a pronounced conformational heterogeneity of myosin-V in all our data sets and characterize it in detail based on 18 high-resolution subset structures.

## Results and discussion
### High-resolution cryo-EM structures of the actomyosin-V complex

To provide insights into the structural transitions of myosin along its motor cycle, we determined the structure of the actomyosin-V complex in three different nucleotide states using single-particle cryo-EM. Specifically, we have decorated aged ADP-bound F-actin (rabbit skeletal α-actin) stabilized by PHD (*Lynen and Wieland, 1938*) with myosin-Va –S1 fragment bound to one essential light chain, hereafter referred to as myosin-V. The complex, referred to as aged actomyosin-V, was either prepared in the absence of a nucleotide or after brief incubation of myosin with Mg$^{2+}$-ADP or Mg$^{2+}$-AppNHp (see Materials and methods for details). AppNHp, also known as AMPPNP, is an ATP analog that has been shown to be non-hydrolyzable by myosin-V (*Yengo et al., 2002*). It is coordinated similarly to ATP in crystal structures of myosin-II (*Bauer et al., 2000*; *Gulick et al., 1997*) and has also been reported to lead to a mixture of a pre- and post-powerstroke conformations in myosin-V (*Yengo et al., 2002*; *Volkmann et al., 2005*). These results suggest that AppNHp can potentially mimic both ATP and ADP-P$_i$ and is thus well suited to capture short-lived actin-bound transition states, such as the weakly bound PPS and post-rigor states (*Sweeney and Houdusse, 2010*).

We collected cryo-EM data sets of the different samples (*Table 1*) and processed them using the helical processing pipeline implemented in the SPHIRE package (*Moriya et al., 2017*; *Pospich et al.,*

**Table 2.** Statistics of aged actomyosin in the strong-ADP state.

Refinement and model building statistics of aged F-actin-PHD in complex with myosin-V in the strong-ADP state.

| | Strong-ADP state: aged F-actin-PHD + myosin-Va-LC + Mg$^{2+}$-ADP | | | | | | | |
|---|---|---|---|---|---|---|---|---|
| | Central 3er/2er | Central 1er (subtracted) | Class 2 | Class 3 | Class 4 | Class 5 | Class 6 | Class 7 |
| **3D refinement statistics** | | | | | | | | |
| Number of helical segments | 871,844 | 871,844 | 140,383 | 107,848 | 113,766 | 107,961 | 118,875 | 104,552 |
| Resolution (Å) | 3.0 | 3.1 | 3.5 | 3.5 | 3.7 | 3.6 | 3.6 | 3.7 |
| Map sharpening factor (Å$^2$) | –60 | –60 | –78 | –78 | –94 | –86 | –83 | –88 |
| **Atomic model statistics** | | | | | | | | |
| Non-hydrogen atoms | 23,334 | 10,171 | 10,149 | 10,149 | 10,086 | 10,066 | 10,113 | 10,139 |
| Cross-correlation masked | 0.85 | 0.83 | 0.83 | 0.83 | 0.80 | 0.82 | 0.83 | 0.80 |
| MolProbity score | 1.35 | 1.23 | 1.28 | 1.36 | 1.38 | 1.36 | 1.35 | 1.39 |
| Clashscore | 6.28 | 4.55 | 5.31 | 6.45 | 6.94 | 6.50 | 6.37 | 7.15 |
| EMRinger score* | 3.42/2.83 | 3.56/3.36 | 3.44/3.49 | 2.83/2.92 | 2.67/2.23 | 2.99/2.92 | 2.92/2.52 | 2.68/2.38 |
| Bond RMSD (Å) | 0.012 | 0.005 | 0.004 | 0.005 | 0.005 | 0.004 | 0.006 | 0.008 |
| Angle RMSD (°) | 1.07 | 0.83 | 0.85 | 0.89 | 0.92 | 0.88 | 0.93 | 1.06 |
| Rotamer outliers (%) | 0.04 | 0.09 | 0.09 | 0.09 | 0.09 | 0.09 | 0.09 | 0.09 |
| Ramachandran favored (%) | 99.65 | 99.68 | 99.76 | 99.68 | 99.84 | 99.84 | 99.84 | 99.84 |
| Ramachandran outliers (%) | 0.00 | 0.00 | 0.00 | 0.00 | 0.00 | 0.00 | 0.00 | 0.00 |
| CaBLAM outliers (%) | 0.7 | 0.9 | 1.1 | 1.3 | 1.2 | 1.0 | 0.8 | 1.4 |

*Values correspond to score against the post-refined map used for real-space refinement/a map filtered to local resolution.

2021; *Stabrin et al., 2020*), which applies helical restraints but no symmetry. For each data set, two all-particle density maps were reconstructed (*Figure 1—figure supplement 1*, see Materials and methods for details). In this way, we achieved nominal resolutions of 3.0 Å/3.1 Å (ADP), 3.2 Å/3.3 Å (rigor), and 2.9 Å/2.9 Å (AppNHp), respectively (*Figure 1—figure supplement 1* and *Figure 1—figure supplement 2*, *Table 2*, *Table 3*, *Table 4*), allowing us to reliably model each state and analyze its molecular interactions.

## Varying conformations in the strong-ADP state of different myosins

The structure of F-actin decorated with myosin-V in complex with Mg$^{2+}$-ADP represents the strong-ADP state, which has high affinity for both F-actin and ADP and directly precedes the nucleotide-free rigor state within the myosin motor cycle. The overall structure encompasses all hallmarks of the strong-ADP state including a closed actin-binding cleft, which allows strong binding to F-actin, and a post-powerstroke lever arm orientation (*Figure 1*, *Figure 1—video 1*), in line with an earlier medium-resolution structure of the same complex (*Wulf et al., 2016*).

The density corresponding to Mg$^{2+}$-ADP is pronounced, indicating high to complete saturation of the active site (*Figure 1*, *Figure 1—figure supplement 3*). The β-phosphate of ADP is tightly coordinated by the P-loop (aa 164–168) via a conserved Walker-A nucleotide binding motif (*Walker et al., 1982*), which is also found in other ATPases as well as G-proteins (*Kull and Endow, 2013*; *Vale, 1996*). The HF helix (aa 169–183) and switch I (aa 208–220) mediate additional contacts by either directly binding to the β-phosphate or coordinating the Mg$^{2+}$ ion (*Figure 1B and C*). The third key loop of the

**Table 3.** Statistics of aged actomyosin in the rigor state.

Refinement and model building statistics of aged F-actin-PHD in complex with myosin-V in the rigor state.

| | Rigor state: aged F-actin-PHD + myosin-Va-LC | | | | |
|---|---|---|---|---|---|
| | Central 3er/2er | Central 1er (subtracted) | Class 1 | Class 2 | Class 4 |
| **3D refinement statistics** | | | | | |
| Number of helical segments | 299,784 | 299,784 | 94,077 | 102,818 | 81,757 |
| Resolution (Å) | 3.2 | 3.3 | 3.5 | 3.5 | 3.6 |
| Map sharpening factor (Å$^2$) | –81 | –80 | –89 | –89 | –87 |
| **Atomic model statistics** | | | | | |
| Non-hydrogen atoms | 23,288 | 10,148 | 10,139 | 10,139 | 10,139 |
| Cross-correlation masked | 0.83 | 0.86 | 0.83 | 0.82 | 0.81 |
| MolProbity score | 1.28 | 1.18 | 1.24 | 1.25 | 1.31 |
| Clashscore | 5.25 | 3.97 | 4.66 | 4.81 | 5.75 |
| EMRinger score* | 3.14/3.39 | 3.41/3.10 | 2.97/3.00 | 3.53/3.00 | 3.01/3.06 |
| Bond RMSD (Å) | 0.005 | 0.014 | 0.005 | 0.005 | 0.005 |
| Angle RMSD (°) | 0.84 | 1.14 | 0.80 | 0.84 | 0.82 |
| Rotamer outliers (%) | 0.04 | 0.00 | 0.00 | 0.00 | 0.00 |
| Ramachandran favored (%) | 99.86 | 99.84 | 99.60 | 99.76 | 99.76 |
| Ramachandran outliers (%) | 0.00 | 0.00 | 0.00 | 0.00 | 0.00 |
| CaBLAM outliers (%) | 0.8 | 0.9 | 0.9 | 0.9 | 0.8 |

*Values correspond to score against the post-refined map used for real-space refinement/a map filtered to local resolution.

active site, switch II (aa 439–448), does not directly contribute to the binding of Mg$^{2+}$-ADP, which is in agreement with its proposed role in ATP hydrolysis and the subsequent release of the inorganic phosphate (*Sweeney et al., 2020*). Yet, switch II contributes to the stability of the active site by forming a hydrogen bond with the HF helix (D437-T170, predicted by PDBsum; *Laskowski et al., 2018*). In addition to the coordination of the β-phosphate, ADP binding is mediated by primarily hydrophobic interactions of the adenosine moiety with the purine-binding loop (*Bloemink et al., 2020*) (aa 111–116)—for brevity, hereafter referred to as A-loop (adenosine-binding loop) (*Figure 1B and C*, *Figure 1—video 1*, *Figure 1—figure supplement 3*). A tyrosine (Y119) trailing the A-loop forms another putative hydrogen bond with the adenosine, completing the coordination of ADP.

The coordination of Mg$^{2+}$-ADP in our structure closely resembles the ones reported for the strong-ADP state of myosin-IB (*Mentes et al., 2018*), myosin-VI (*Gurel et al., 2017*), and myosin-XV (*Gong et al., 2021*; *Figure 1—figure supplement 4*). Only the position of switch I differs appreciably between myosins, ultimately resulting in varying positions of the coordinated Mg$^{2+}$ ion. These differences highlight that while the general architecture of the active site is common to all myosins, small local reorganizations occur and possibly account for the different kinetics within the myosin superfamily. In contrast to the similarities of the active site, the overall structures of the strong-ADP states of myosin-V, -IB, and -XV differ considerably, resulting in lever arm orientations deviating by 71° and 22°, respectively (*Figure 1—figure supplement 5*).

## Structural transition of myosin-V upon ADP release

The structure of the actomyosin-V complex in the absence of any nucleotide in myosin represents the rigor state (*Figure 2*, *Figure 2—video 1*). In addition to an unoccupied and open active site (*Figure 2—video 1*, *Figure 1—figure supplement 3*), the actin-binding cleft is closed, facilitating strong binding to F-actin, and the lever arm adopts a post-powerstroke orientation (*Figure 2*, *Figure 2—video 1*). These features are common to all rigor structures solved to date (*Banerjee et al., 2017*; *Behrmann*

**Table 4.** Statistics of aged actomyosin in the post-rigor transition (PRT) state.
Refinement and model building statistics of aged F-actin-PHD in complex with myosin-V in the PRT state (bound to AppNHp).

| | Post-rigor transition state: aged F-actin-PHD + myosin-Va-LC + Mg²⁺-AppNHp | | | | | | | |
| --- | --- | --- | --- | --- | --- | --- | --- | --- |
| | Central 3er/2er | Central 1er (subtracted) | Class 1 | Class 3 | Class 4 | Class 5 | Class 6 | Class 8 |
| **3D refinement statistics** | | | | | | | | |
| Number of helical segments | 2,446,218 | 2,446,218 | 330,197 | 365,722 | 350,069 | 321,218 | 277,487 | 343,500 |
| Resolution (Å) | 2.9 | 2.9 | 3.4 | 3.3 | 3.4 | 3.3 | 3.4 | 3.3 |
| Map sharpening factor (Å²) | −80 | −100 | −113 | −106 | −114 | −106 | −111 | −104 |
| | | | | | | | | |
| **Atomic model statistics** | | | | | | | | |
| Non-hydrogen atoms | 23,370 | 10,189 | 10,125 | 10,189 | 10,154 | 10,189 | 10,085 | 10,189 |
| Cross-correlation masked | 0.85 | 0.84 | 0.84 | 0.86 | 0.85 | 0.85 | 0.83 | 0.84 |
| MolProbity score | 1.25 | 1.15 | 1.17 | 1.24 | 1.20 | 1.26 | 1.37 | 1.18 |
| Clashscore | 4.76 | 3.56 | 3.78 | 4.64 | 4.12 | 4.99 | 6.74 | 3.95 |
| EMRinger score* | 3.29/3.45 | 3.82/3.40 | 3.35/3.07 | 3.58/3.45 | 3.18/3.35 | 2.94/2.97 | 3.01/3.01 | 3.09/2.88 |
| Bond RMSD (Å) | 0.004 | 0.012 | 0.009 | 0.014 | 0.009 | 0.014 | 0.009 | 0.005 |
| Angle RMSD (°) | 0.78 | 1.01 | 0.96 | 1.15 | 0.97 | 1.14 | 1.08 | 0.81 |
| Rotamer outliers (%) | 0.08 | 0.09 | 0.09 | 0.09 | 0.09 | 0.09 | 0.09 | 0.09 |
| Ramachandran favored (%) | 99.86 | 99.84 | 99.84 | 99.84 | 99.84 | 99.84 | 99.84 | 99.84 |
| Ramachandran outliers (%) | 0.00 | 0.00 | 0.00 | 0.00 | 0.00 | 0.00 | 0.00 | 0.00 |
| CaBLAM outliers (%) | 1.1 | 1.2 | 1.1 | 1.1 | 1.2 | 1.2 | 1.4 | 0.7 |

*Values correspond to score against the post-refined map used for real-space refinement/a map filtered to local resolution.

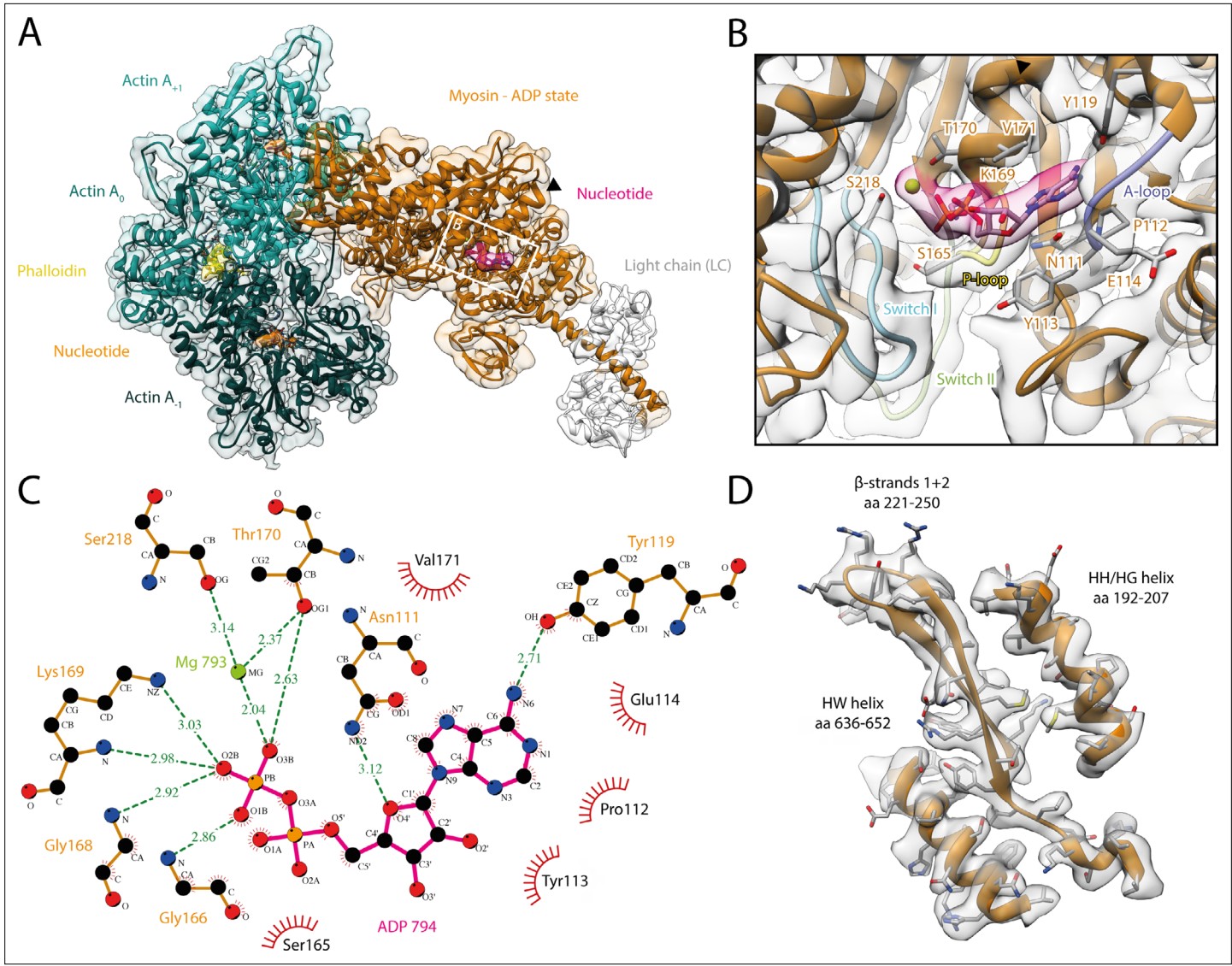

**Figure 1.** Structure and active site of the aged actomyosin-V complex bound to ADP. (**A**) Atomic model and LAFTER density map of the central myosin-V-LC subunit (orange, LC: white) bound to aged F-actin-PHD (shades of sea green, three subunits shown, $A_{-1}$ to $A_{+1}$). Nucleotides and PHD are highlighted in orange, pink, and yellow, respectively. The HF helix is marked by a black arrowhead. (**B**) Close-up view of the myosin active site consisting of the P-loop (yellow, 164–168), switch I (blue, aa 208–220), switch II (green, aa 439–448), and the A-loop (purple, aa 111–116). Only side chains involved in the binding of ADP are displayed, also see *Figure 1—figure supplement 3*. (**C**) 2D protein-ligand interaction diagram illustrating the coordination of $Mg^{2+}$-ADP by hydrogen bonds (dashed green lines) and hydrophobic interactions (red rays). (**D**) Illustration of the model-map agreement within a central section of myosin. Most side chains are resolved by the post-refined density map (transparent gray). See *Figure 1—video 1* for a three-dimensional visualization and *Figure 1—figure supplements 1–2* for an overview of the processing pipeline and the cryo-EM data, respectively. A comparison of the strong-ADP state of different myosins can be found in *Figure 1—figure supplements 4 and 5*. *Figure 1—figure supplement 6* illustrates the domain architecture of myosin.

The online version of this article includes the following video and figure supplement(s) for figure 1:

**Figure supplement 1.** Schematic of the cryo-EM processing pipeline.

**Figure supplement 2.** Overview of the cryo-EM data and resolution of aged F-actin-PHD in complex with myosin-V in the rigor, ADP, and AppNHp state.

**Figure supplement 3.** Nucleotide densities at and organization of the active sites of actin and myosin.

**Figure supplement 4.** Comparison of the coordination of $Mg^{2+}$-ADP in different myosins.

**Figure supplement 5.** Structural variations of the rigor and strong-ADP states of different actomyosin complexes.

**Figure supplement 6.** Domain architecture of the myosin motor domain.

*Figure 1 continued on next page*

*Figure 1 continued*

**Figure 1—video 1.** Structure of the aged actomyosin-V complex bound to ADP.
https://elifesciences.org/articles/73724/figures#fig1video1

*et al., 2012*; *Doran et al., 2020*; *Fujii and Namba, 2017*; *Gong et al., 2021*; *Gurel et al., 2017*; *Mentes et al., 2018*; *Risi et al., 2021*; *Robert-Paganin et al., 2021*; *Vahokoski et al., 2020*; *von der Ecken et al., 2016*). Yet, the structures of different myosins vary, particularly in the orientation of the lever arm (*Figure 1—figure supplement 5A*).

While the actomyosin interface of the rigor state of myosin-V is basically indistinguishable from the one in the strong-ADP state, the lever arm orientations of the two states differ by ~9° (*Figure 3A*, *Figure 3—video 1A*), in agreement with a previously reported rotation of 9.5° (*Wulf et al., 2016*). The overall architecture of our rigor state structure not only is in good agreement with the medium-resolution cryo-EM structure published earlier (*Wulf et al., 2016*), but also strongly resembles the rigor-like crystal structures solved for this myosin isoform (*Figure 2C–F*; *Coureux et al., 2003*; *Coureux et al., 2004*).

As the strong-ADP and rigor state represent sequential states within the myosin motor cycle, a comparison of the respective high-resolution structures allows the detailed description of the structural transition of myosin-V upon $Mg^{2+}$-ADP release (*Figure 3*, *Figure 3—video 1*). In addition to the ~9° lever arm rotation described above (*Figure 3A*), the two sequential states differ primarily in their conformation of the central transducer β-sheet and the N-terminal domain, which twist and rotate, respectively (*Figure 3C–E*, *Figure 3—video 1*; see *Figure 1—figure supplement 6* for an overview of the myosin domain architecture). Notably, the structural changes are not transmitted to the U50 and L50 domains and thus do not alter the actin-binding interface (*Figure 3A and C*, *Figure 3—video 1A and B*).

The transducer rearrangements are directly linked to a reorganization of the active site that accounts for the reduced $Mg^{2+}$-ADP affinity of the rigor state. By promoting a piston movement of the HF helix, twisting of the transducer increases the distance between the P-loop and switch I, thereby opening the active site (*Figure 3B and D*, *Figure 3—video 1A–C*). The resulting conformation is incompatible with the $Mg^{2+}$-coordinating hydrogen bond between the HF helix and switch II (T170-D437). Loss of $Mg^{2+}$ is thought to lead to the weak-ADP state of myosin (*Coureux et al., 2004*), which is so named due to its low nucleotide affinity that promotes the release of ADP. The subsequent rigor state is stabilized by a new network of hydrogen bonds formed between lysine K169 (HF helix, previously coordinated to the β-phosphate of ADP), and aspartate D437 and isoleucine I438 (switch II).

Upon $Mg^{2+}$-ADP release, the A-loop also undergoes a small lateral shift (*Figure 3B, D and E*, *Figure 3—video 1C and D*). In this way, it likely stabilizes the twisting of the transducer and the N-terminal domain rotation. Surprisingly, the role of the A-loop in both the coordination of ADP and the coupling of the active site to the periphery has not been fully appreciated previously, although it is also involved in nucleotide binding in other myosins (*Bloemink et al., 2020*). Given their central importance for the coordination of $Mg^{2+}$-ADP (*Figure 1*), we propose that the P-loop, the A-loop, and switch I contribute to the sensing of the nucleotide state and its transmission from the nucleotide-binding pocket to the periphery. Their mutual interplay defines the orientation of the N-terminal domain relative to the U50 and L50 subdomains. In this way, small changes in the active site (~1–2 Å) are amplified into significant rotations of the N-terminal and converter domain, eventually leading to a lever arm swing of ~9° upon $Mg^{2+}$-ADP release (*Figure 3*, *Figure 3—video 1*).

Our high-resolution structures of the strong-ADP and rigor state are consistent with the sequential release of $Mg^{2+}$ and ADP due to the isomerization of myosin to a conformation with reduced nucleotide affinity. In line with this, ADP binding to the rigor state can favor the reversal of this isomerization in the presence of $Mg^{2+}$.

A similar structural transition upon $Mg^{2+}$-ADP release has been reported for myosin-IB, -V, and -VI based on medium- and high-resolution cryo-EM structures (*Gurel et al., 2017*; *Mentes et al., 2018*; *Wulf et al., 2016*), suggesting a common coupling mechanism. Although most of the details are intriguingly similar, for example, the remodeling of hydrogen bonds due to the piston movement of the HF helix (*Mentes et al., 2018*), we find notable differences in the extent of the lever arm swing associated with $Mg^{2+}$-ADP release (*Figure 1—figure supplement 5*), as well as the conformation of the relay helix, which partially unwinds in myosin-IB and -VI to allow for the larger lever arm swings

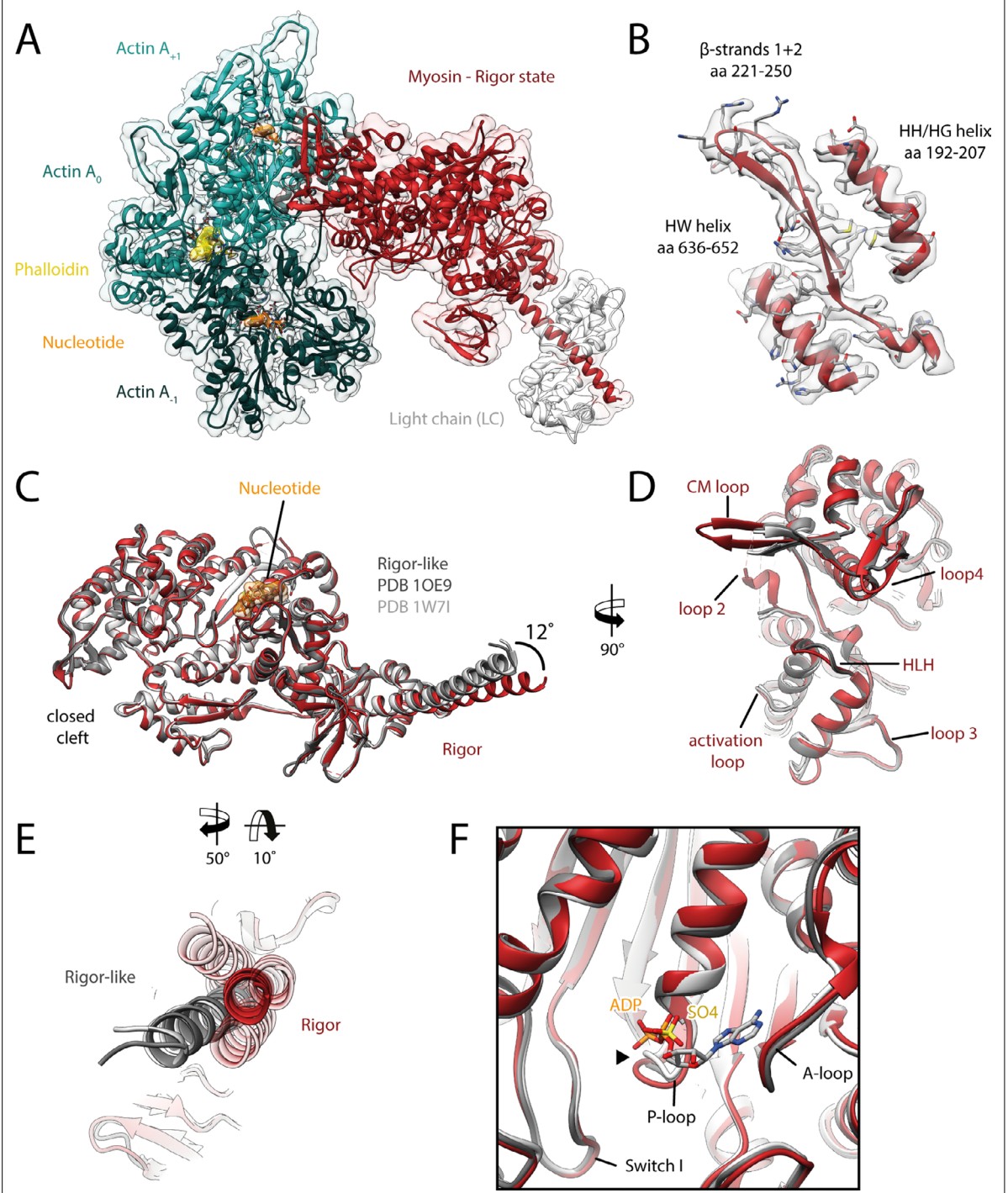

**Figure 2.** Structure of the aged actomyosin-V complex in the rigor state. (**A**) Atomic model and LAFTER density map of the central myosin-V-LC subunit (red, LC: white) bound to aged F-actin-PHD (shades of sea green, three subunits shown, A$_{-1}$ to A$_{+1}$). Nucleotides and PHD are highlighted in orange and yellow, respectively. (**B**) Illustration of the model-map agreement within a central section of myosin. Most side chains are resolved by the post-refined density map (transparent gray). See *Figure 2—video 1* for a three-dimensional visualization. (**C–F**) Comparison of the rigor state of myosin-V with crystal structures of the same myosin in the rigor-like state (PDB: 1OE9; *Coureux et al., 2003*; and PDB: 1W7I, also called weak-ADP state; *Coureux et al., 2004*; shades of gray). (**C**) Superposition of atomic models illustrating that deviations are limited to the actin interface, particularly (**D**) the CM loop, loop 4, and loop 2 and (**E**) the lever arm. Interestingly, the lever arm orientation seen in the rigor-like states does not superimpose with any conformation seen for the rigor complex (average: red; and 3D classes: transparent red), but localizes outside of its conformational space. (**F**) The active site is open in both the rigor and rigor-like states, and the SO$_4$ and ADP bound to the rigor-like crystal structures only give rise to small, isolated changes of the P-loop (highlighted by a black arrowhead). Differences in the rigor-like structure can be readily attributed to the absence of F-actin and crystal packing,

*Figure 2 continued on next page*

*Figure 2 continued*

respectively.

The online version of this article includes the following video for figure 2:

**Figure 2—video 1.** Structure of the aged actomyosin-V complex in the rigor state.

https://elifesciences.org/articles/73724/figures#fig2video1

(*Gurel et al., 2017*; *Mentes et al., 2018*). Interestingly, myosin-IB not only performs a larger lever arm swing (25°) (*Mentes et al., 2018*), but is also almost 40 times more sensitive to force than myosin-V (9° swing) (*Laakso et al., 2008*; *Veigel et al., 2005*). Since load will more easily prevent the isomerization of myosin if $Mg^{2+}$-ADP release requires a large converter swing, we propose that the force sensitivity, which tunes the kinetics of the transition to the rigor state (*Kovács et al., 2007*; *Laakso et al., 2008*; *Takagi et al., 2006*; *Veigel et al., 2005*), increases with the extent of the lever arm swing upon $Mg^{2+}$-ADP release.

## AppNHp gives rise to a strongly bound PRT state

We determined the structure of F-actin-myosin-V in complex with the non-hydrolyzable ATP analog AppNHp with the aim to characterize a potentially short-lived, weakly bound state of myosin. The resulting cryo-EM density map shows strong density for AppNHp, indicating high to complete saturation (*Figure 4*, *Figure 4—video 1*, *Figure 1—figure supplement 3*). Interestingly, the density also suggests the presence of two ions, both likely corresponding to $Mg^{2+}$, given the size of the density and the buffer composition. While one ion occupies approximately the position that $Mg^{2+}$ takes in the active site of the strong-ADP state, namely close to the γ-phosphate of AppNHp, the other one resides in between the α- and β-phosphates of AppNHp (*Figures 1 and 4*).

Similar to ADP, AppNHp is coordinated by a network of hydrogen bonds and additional hydrophobic interactions with the P-loop, switch I, and the A-loop (*Figure 1C* and *Figure 4C*). The details of the interactions, however, differ due to the different sizes of the two nucleotides and their relative positions in the active site, that is, the γ-phosphate of AppNHp almost takes the position of the β-phosphate of ADP relative to the HF helix (*Figure 4*, *Figure 1—figure supplement 3*, *Figure 1*).

Surprisingly, and in contrast to a previous low-resolution cryo-EM reconstruction (*Volkmann et al., 2005*), the overall structure of AppNHp-bound myosin-V is reminiscent of the rigor state (*Figure 4—figure supplement 1*). In particular, myosin is strongly bound to F-actin and adopts a post-powerstroke lever arm orientation (*Figure 4*, *Figure 4—video 1*, *Figure 4—figure supplement 1A and B*). The active site of AppNHp-bound myosin also closely resembles that of the rigor state, and thereby significantly deviates from the conformation found in the strong-ADP state (*Figure 4—figure supplement 1C and D*).

The compatibility of an ATP analog, specifically the presence of a γ-phosphate at the active site, with strong F-actin binding is initially puzzling and seemingly at odds with the reported reciprocal nature of these two processes (*Coureux et al., 2004*; *Kühner and Fischer, 2011*). A comparison of our AppNHp-bound structure with a rigor-like crystal structure of myosin-V with ADP weakly bound to its active site (*Coureux et al., 2004*) resolves this conflict (*Figure 4—figure supplement 2A*). The relative position of AppNHp and ADP in these two structures as well as their coordination, which in particular lacks contacts between K169 of the P-loop and the β-phosphate, is almost identical, suggesting that AppNHp is only weakly bound in our structure and therefore compatible with strong F-actin binding. Interestingly, a similar coordination was observed for $Mg^{2+}$-ADP in a putative strong-ADP to rigor transition state cryo-EM structure of myosin-IB (*Mentes et al., 2018*; *Figure 4—figure supplement 2B*). These comparisons indicate that AppNHp and ADP can both weakly bind to myosin in a conformation reminiscent of the rigor.

Our prior kinetic studies (*De La Cruz et al., 1999*; *Yengo et al., 2002*) demonstrated that AppNHp reduces the binding affinity of myosin-V for F-actin by >5000-fold as compared to the rigor state, thus favoring dissociation. A weakened affinity is also supported by the higher concentrations required to achieve decoration of F-actin with myosin in the AppNHp state (see Materials and methods). AppNHp also induces greater structural flexibility in myosin-V (see below) as compared to the rigor state, which may facilitate the transition to a detached state. Based on the presented structural and prior kinetic studies, we propose that our AppNHp-bound myosin-V structure represents a post-rigor transtion

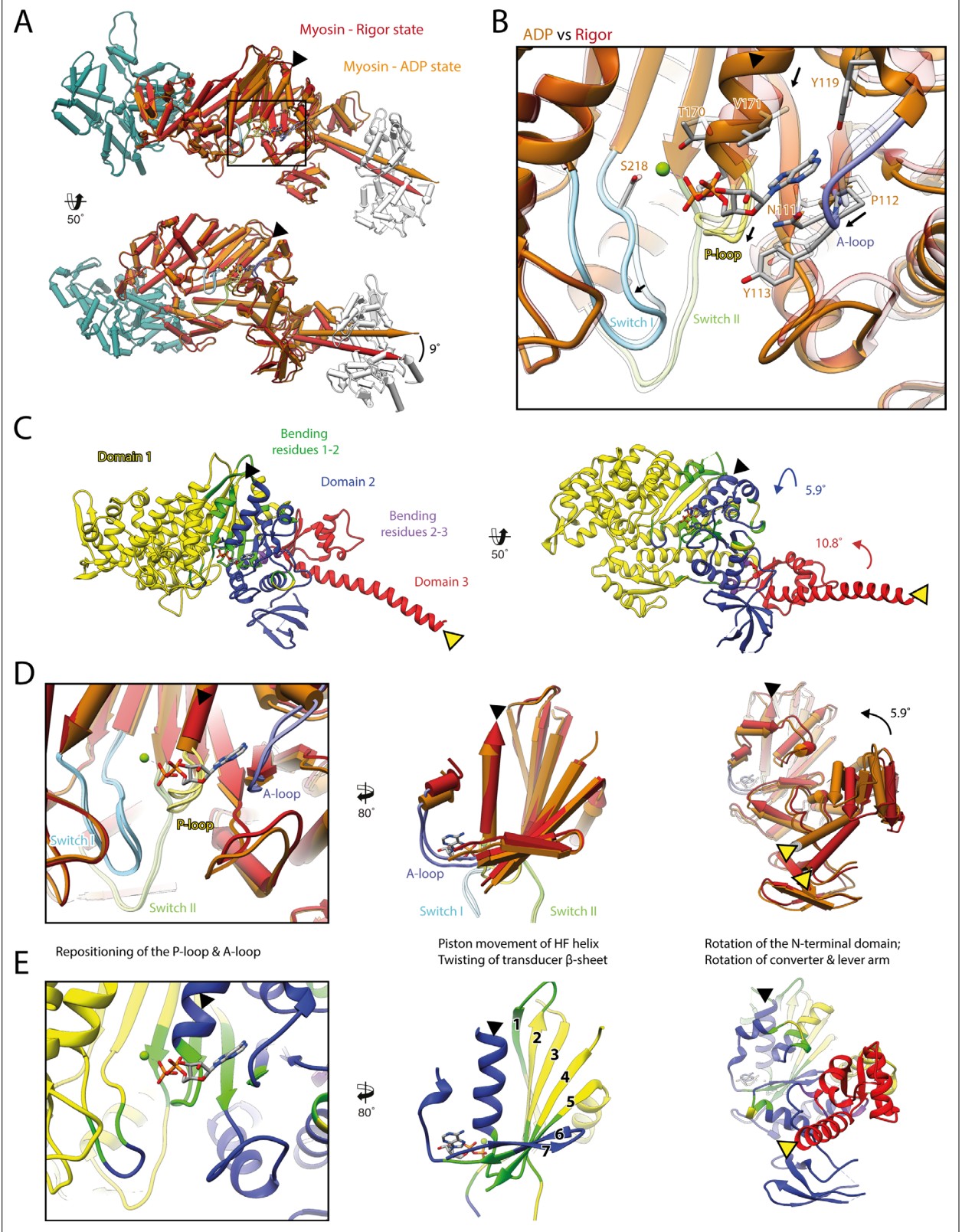

**Figure 3.** Structural transition of myosin-V upon $Mg^{2+}$-ADP release. (**A**) Superposition of the strong-ADP (orange) and rigor (red) atomic models. Changes at the active site (black box) are not transmitted to the actomyosin interface, but to the N-terminal and converter domain, resulting in a lever swing of 9°. (**B**) Close-up view of the active site showing the structural rearrangements upon $Mg^{2+}$-ADP release (indicated by black arrows). The rigor structure is shown as transparent; see *Figure 1* for color code. (**C**) Illustration of domain movements associated with $Mg^{2+}$-ADP release predicted by

*Figure 3 continued on next page*

*Figure 3 continued*

DynDom (*Hayward and Lee, 2002*). Identified domains correlate well with myosins structural domains (see *Figure 1—figure supplement 6*) with domain 1 (yellow, 452 residues), domain 2 (181 residues, blue), and domain 3 (93 residues, red) representing the L50 and U50 domains, the N-terminal domain, and the converter domain, respectively. Bending residues primarily localize to the P-loop, the A-loop, and the central transducer β-sheet (1–2, green), as well as to a small part of the N-terminal and converter domain (2–3, purple). (**D**) Scheme illustrating the structural changes associated with Mg$^{2+}$-ADP release. (**E**) Same views as in (**D**), but colored by DynDom domains, also see (**C**). The HF helix and the lever arm are highlighted by a black and a yellow arrowhead, respectively. Models were aligned on F-actin. See *Figure 3—video 1* for a three-dimensional visualization.

The online version of this article includes the following video for figure 3:

**Figure 3—video 1.** Structural transition of myosin-V upon ADP release.

https://elifesciences.org/articles/73724/figures#fig3video1

(PRT) state that allows to visualize how ATP binds in the rigor state, prior to the transition that involves a switch I movement and promotes detachment of myosin from F-actin. The characteristic weak coordination of AppNHp in the PRT state allows myosin to remain strongly bound to F-actin until a strong coordination of the nucleotide is established. The report of a transition state with weakly bound ADP (*Mentes et al., 2018*; *Figure 4—figure supplement 2B*) suggests that weak nucleotide binding is a common scheme and that the PRT state is therefore not limited to AppNHp. The visualization of an ATP analog bound to a state reminiscent of the rigor shows that ATP mainly binds via its adenine ring, as does ADP (*Figure 1*). It also explains how the γ-phosphate can fit into the relatively small pocket created by the rigor conformation of the P-loop (*Figure 4*), and how its presence leads to local changes of the active site facilitating a tight coordination (*Figure 4—figure supplement 1*). In this way, the PRT state provides new insights on how myosin detaches from F-actin and indicates that the theoretical weakly bound post-rigor state (*Sweeney and Houdusse, 2010*; *Walklate et al., 2016*) is unlikely to be populated within the motor cycle.

Although we find myosin-V-AppNHp strongly bound to F-actin in the PRT state (*Figure 4—figure supplement 1*), we had to significantly increase the myosin concentration to achieve full decoration of actin filaments (see Materials and methods for details), in agreement with a weaker binding affinity (*Konrad and Goody, 2005*; *Yengo et al., 2002*). We therefore conclude that AppNHp can potentially lead to different structural states, similar to ADP in myosin-IB (*Mentes et al., 2018*). Likely due to large differences in the binding affinity of these states or rapid detachment of myosin from F-actin, we only find myosin bound to F-actin in the PRT state. In line with this assumption, we find a significant amount of unbound myosin in the background of our AppNHp data sets (*Figure 4—figure supplement 3*). The 3D reconstruction and thus identification of the structural state of the background myosin were unfortunately impeded by a strong orientational preference of the myosin particles (*Figure 4—figure supplement 3B*). Further studies are therefore required to test the conformation of AppNHp-bound myosin-V in absence of F-actin.

## Conservation and specificity of the actomyosin-V interface

A comparison of the three states of the actomyosin-V complex (strong-ADP, rigor, and PRT state) reveals a striking similarity of the actomyosin interface (*Figure 5*, *Figure 5—video 1*). The atomic models superimpose almost perfectly with only little variations in the orientation of some incompletely resolved side chains. The remarkable similarity suggests that the same set of interactions is maintained during all strongly bound states of the myosin motor cycle, despite their varying F-actin-binding affinities. Differences in the affinity might therefore not be linked to altered contacts, but rather to the degree of structural flexibility inherent to each state (see below).

The actomyosin-V interface comprises six structural elements, namely the cardiomyopathy (CM) loop (aa 376–392), loop 4 (aa 338–354), the helix-loop-helix (HLH) motif (505–531), the activation loop (aa 501–504), loop 3 (aa 532–546), and loop 2 (aa 594–635) (*Figure 5*, *Figure 5—video 1*). While these elements represent a common set of actin-binding elements, most of which have conserved hydrophobic and electrostatic properties, not all myosins utilize all of them. Moreover, the precise nature of individual interactions and the residues involved varies considerably among myosins, largely due to sequence variations known to tune the kinetic properties of myosin (*Mentes et al., 2018*; *Robert-Paganin et al., 2021*). Comparisons of the actomyosin interface of different myosins are therefore essential for identifying common and specific features of the myosin superfamily.

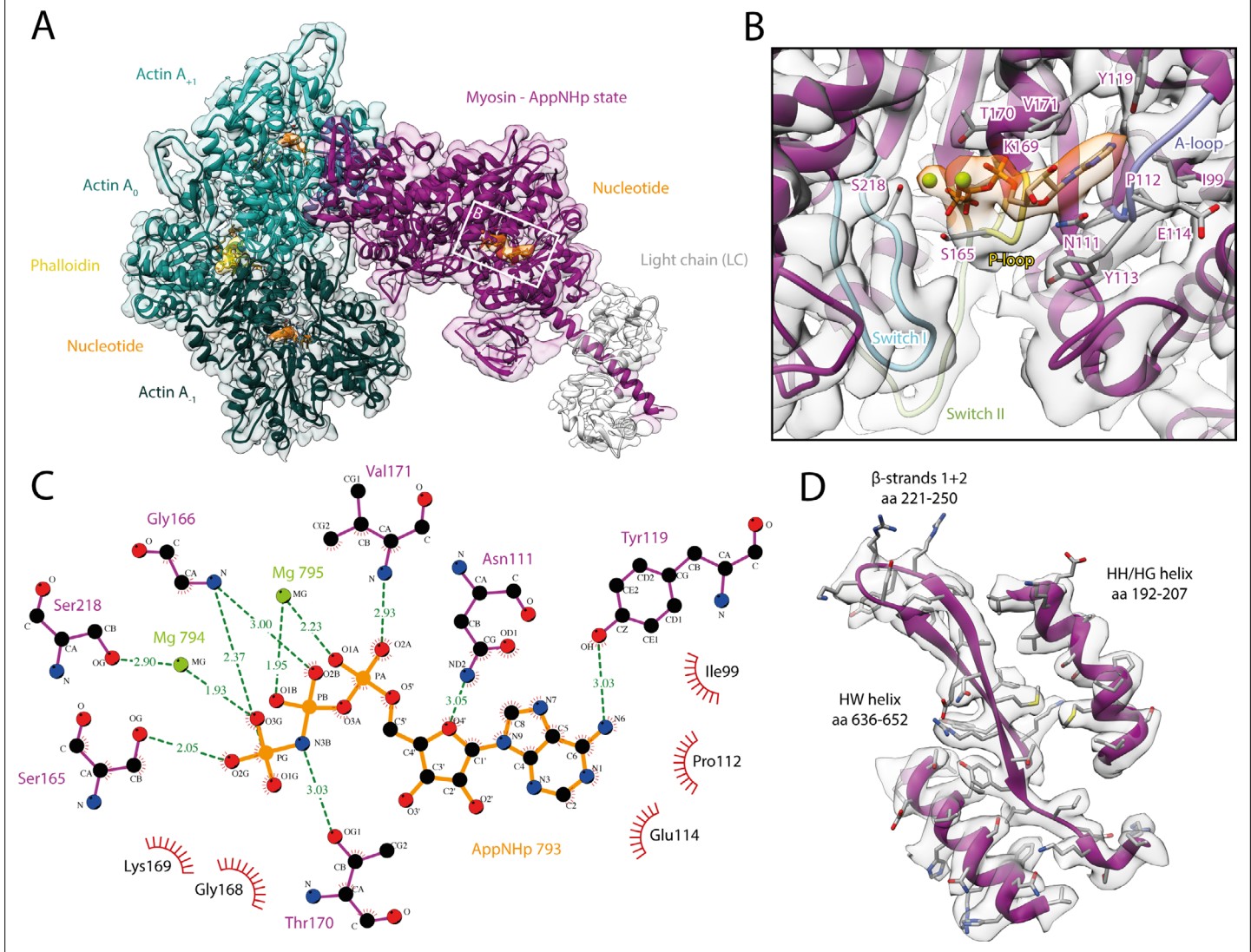

**Figure 4.** Structure and active site of the aged actomyosin-V complex bound to AppNHp. (**A**) Atomic model and LAFTER density map of the central myosin-V-LC subunit (purple, LC: white) bound to aged F-actin-PHD (shades of sea green, three subunits shown, A$_{-1}$ to A$_{+1}$). Nucleotides and PHD are highlighted in orange and yellow, respectively. (**B**) Close-up view of the myosin active site; see *Figure 1* for color code. Only side chains involved in the binding of AppNHp are displayed. The density suggests the presence of two Mg$^{2+}$ ions coordinating the γ, and α- and β-phosphate, respectively; also see *Figure 1—figure supplement 3* and *Figure 4—video 1*. (**C**) 2D protein-ligand interaction diagram illustrating the coordination of Mg$^{2+}$-AppNHp by hydrogen bonds (dashed green lines) and hydrophobic interactions (red rays). (**D**) Illustration of the model-map agreement within a central section of myosin. Most side chains are resolved by the post-refined density map (transparent gray). See *Figure 4—figure supplements 1–3* for comparisons of the AppNHp-myosin-V structure with other structures as well as an analysis of unbound myosin in the AppNHp data set.

The online version of this article includes the following video and figure supplement(s) for figure 4:

**Figure supplement 1.** Comparison of AppNHp-bound myosin-V with myosin-V in the rigor and strong-ADP state.

**Figure supplement 2.** Common active site conformation in transition states with weakly bound nucleotide.

**Figure supplement 3.** 2D cryo-EM data of unbound myosin-V in complex with AppNHp.

**Figure 4—video 1.** Structure of the aged actomyosin-V complex bound to AppNHp.

https://elifesciences.org/articles/73724/figures#fig4video1

A detailed comparison of the actomyosin interface of myosin-V with previously published actomyosin structures (*Banerjee et al., 2017*; *Behrmann et al., 2012*; *Doran et al., 2020*; *Gong et al., 2021*; *Gurel et al., 2017*; *Mentes et al., 2018*; *Risi et al., 2021*; *Robert-Paganin et al., 2021*; *Vahokoski et al., 2020*; *von der Ecken et al., 2016*) shows many common features, but also some myosin-V-specific ones. The tightest and most conserved contact is formed by the HLH motif (*Robert-Paganin*

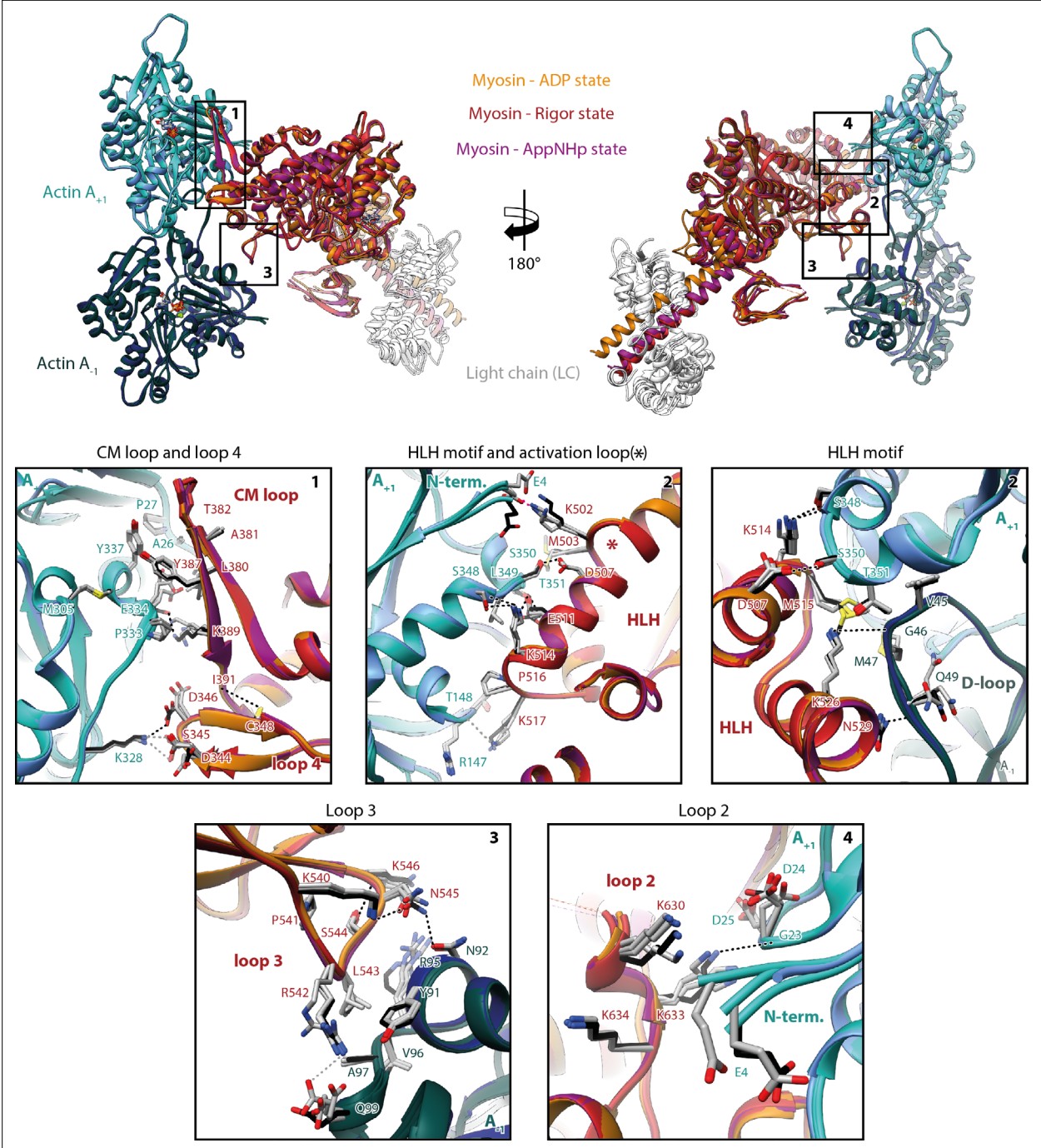

**Figure 5.** Indistinguishable actomyosin interfaces in the strong-ADP, rigor, and post-rigor transition (PRT) state. Comparison of the actomyosin-V interface within all three states (rigor: red; strong-ADP: orange; and AppNHp-bound PRT: purple) illustrating the remarkable similarity of interactions with F-actin. (Top) Front and back views of the central myosin molecule and the two actin subunits it is bound to (shades of green and blue, $A_{+1}$ and $A_{-1}$; see *Figure 1—figure supplement 3* for color code). Black boxes indicate the location of close-up views shown below. (Bottom) Close-up views of all actin-myosin interfaces including the cardiomyopathy (CM) loop, the helix-loop-helix (HLH) motif, loops 2–4, and the activation loop (highlighted by an asterisk). Side chains of key residues are displayed and labeled for all states (rigor: black; ADP and AppNHp: gray). Dashed lines indicate hydrogen bonds predicted for the rigor (black) and ADP/AppNHp state (gray), respectively. See *Figure 5—video 1* for a three-dimensional visualization including density maps.

The online version of this article includes the following video for figure 5:

**Figure 5—video 1.** Conservation of the actomyosin-V interface.

https://elifesciences.org/articles/73724/figures#fig5video1

*et al., 2021*). In analogy to other myosins, it relies primarily on extensive hydrophobic contacts with F-actin, complemented by a series of hydrogen bonds (predicted by PDBsum [*Laskowski et al., 2018*], *Figure 5*, *Figure 5—video 1E and F*). The comparably short CM loop of myosin-V is also highly conserved, with respect to its hydrophobic nature. However, unlike the CM loop of other myosins (*Fujii and Namba, 2017*; *Gurel et al., 2017*; *Mentes et al., 2018*; *Risi et al., 2021*; *von der Ecken et al., 2016*), its tip does not engage in complementary electrostatic interactions (*Figure 5*, *Figure 5—video 1C*). The conformation we found for loop 4 differs from all others reported so far. Not only is it more compact, folding in a β-hairpin, but it also localizes closer to the base of the CM loop, where it is stabilized by a non-conserved hydrogen bond between C348 and I391 (*Figure 5*, *Figure 5—video 1C*). However, its electrostatic interactions with F-actin are reminiscent of those reported for other myosins (*Fujii and Namba, 2017*; *Gurel et al., 2017*; *Risi et al., 2021*; *von der Ecken et al., 2016*). Loop 2 is exceptionally long in myosin-V and only partially resolved in our structures (*Figure 5—video 1*). While this is also the case for most actomyosin structures resolved so far (*Banerjee et al., 2017*; *Doran et al., 2020*; *Gong et al., 2021*; *Risi et al., 2021*; *Robert-Paganin et al., 2021*; *von der Ecken et al., 2016*), loop 2 of myosin-V stands out by the unique α-helical fold of its C-terminal part (*Figure 5—video 1I*). This fold facilitates a compact packing of basic residues and thereby promotes the electrostatic interactions commonly found at the loop 2 interface. The activation loop is a structural element that does not contribute to F-actin binding in all myosins (*Gurel et al., 2017*; *Robert-Paganin et al., 2021*). In myosin-V, it forms primarily electrostatic interactions with the N-terminus of F-actin, but does not lead to its ordering, as has been reported for other myosins (*Figure 5*, *Figure 5—video 1E*; *Banerjee et al., 2017*; *Behrmann et al., 2012*; *Fujii and Namba, 2017*; *Mentes et al., 2018*; *Vahokoski et al., 2020*). The last structural element involved in actin binding is loop 3. It forms the so-called Milligan contact (*Milligan et al., 1990*), which is strong in myosin-V and includes electrostatic and hydrophobic interactions as well as several hydrogen bonds (*Figure 5*, *Figure 5—video 1H*). The contact is furthermore strengthened by hydrogen bonds between K540-N545 and S544-K546 that stabilize the conformation of loop 3. Interestingly, a strong Milligan contact has also been reported for myosin-IB and -VI (*Gurel et al., 2017*; *Mentes et al., 2018*), whereas no or only weak interactions were found in class II myosins (*Doran et al., 2020*; *Fujii and Namba, 2017*; *Risi et al., 2021*; *von der Ecken et al., 2016*). We therefore speculate that an intimate Milligan contact might be a general feature of myosins with

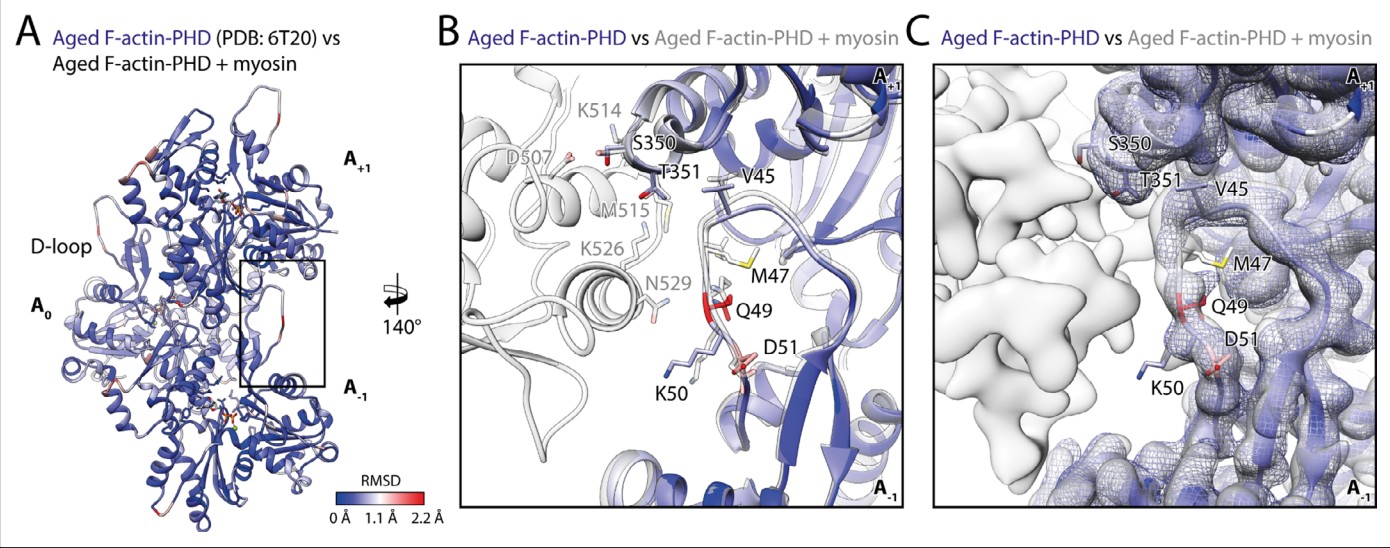

**Figure 6.** Myosin binding gives rise to subtle structural changes of aged PHD-stabilized F-actin. Illustration of the structural similarity of aged F-actin-PHD in the absence and presence of myosin. (**A**) Atomic model of aged F-actin-PHD (PDB: 6T20; *Pospich et al., 2020*; three subunits shown, $A_{-1}$ to $A_{+1}$) color-coded by the backbone root mean square deviation (RMSD) of this structure with the one of aged F-actin-PHD decorated with myosin-V in the rigor state. (**B**) Close-up view of the D-loop interface illustrating that the structural changes associated with myosin binding are small. For a direct comparison, the atomic model of the rigor actomyosin-V complex is superimposed (transparent gray). F-actin subunits were aligned individually to account for errors in the calibration of the pixel size. (**C**) Comparison of LAFTER density maps of aged F-actin-PHD on its own (blue mesh) and bound to myosin-V (gray). For guidance, the atomic model of F-actin-PHD colored by RMSD is also shown. See Table 5 for a comparison of helical symmetry parameters.

**Table 5.** Summary of helical symmetry parameters.

Overview of helical symmetry parameters of aged PHD-stabilized and young JASP-stabilized actomyosin-V complexes. For a direct comparison, the parameters of aged F-actin-PHD (PDB: 6T20; *Pospich et al., 2020*) and young F-actin-JASP (PDB: 5OOD; *Merino et al., 2018*) are shown alongside. Differences in both the helical rise and twist can be readily explained by errors of the pixel size, which is not identical for all data sets. Helical parameters were estimated from the atomic model of five consecutive subunits independently fitted into the map; see *Pospich et al., 2017* for details. To make results more comparable, only actin subunits were considered during fitting. Note that fitting inaccuracies can also give rise to small deviations.

|  | Rise (Å) | Twist (°) | Pixel size (Å) |
|---|---|---|---|
| Helical symmetry |  |  |  |
| Aged F-actin-PHD+ rigor | 27.82±0.02 | −167.27±0.02 | 1.06 |
| Aged F-actin-PHD+ ADP | 27.81±0.02 | −167.32±0.02 | 1.06 |
| Aged F-actin-PHD+ AppNHp | 27.77±0.02 | −167.32±0.02 | 1.10 |
| Aged F-actin-PHD (PDB: 6T20) | 27.59±0.02 | −166.9±0.1 | 1.14 |
| Young F-actin-JASP | 27.85±0.08 | −166.87±0.02 | 1.10 |
| Young F-actin-JASP+ Rigor | 27.72±0.01 | −167.06±0.02 | 1.10 |
| Young F-actin-JASP (PDB: 5OOD) | 27.39 | −166.41 | 1.09 |

long actin-attachment lifetimes and high binding affinities for F-actin and ADP, allowing them to bind particularly tightly to fulfill their function as cargo transporters or molecular anchors.

In summary, we demonstrated that myosin-V establishes a maximum of contacts with F-actin, utilizing all six potential binding elements (*Figure 5*, *Figure 5—video 1E and F*). In addition, we have identified a previously unseen α-helical fold of the C-terminus of loop 2 (*Figure 5*, *Figure 5—video 1I*), which possibly strengthens the interactions at this interface.

## Myosin-V specifically selects the closed D-loop conformation of F-actin

To assess the structural effect of myosin binding on F-actin, we compared the structure of aged F-actin-PHD in the presence (rigor state, representative for all states) and absence of myosin-V (PDB: 6T20; *Pospich et al., 2020*; *Figure 6*). The observed differences are subtle and primarily involve the DNase-binding loop (D-loop, aa 39–55) of F-actin and loops known for their flexibility (*Pospich et al., 2020*). The most prominent alteration involves glutamine Q49 within the D-loop, which moves away from the actomyosin interface by ~2 Å to enable the formation of a hydrogen bond with N529 in the HLH motif of myosin (*Figure 5* and *Figure 6*). Similar, but not identical, subtle changes have been reported for other actomyosins (*Behrmann et al., 2012*; *Gong et al., 2021*; *Gurel et al., 2017*; *Robert-Paganin et al., 2021*; *von der Ecken et al., 2016*), in addition to an ordering of the N-terminus of actin (*Banerjee et al., 2017*; *Behrmann et al., 2012*; *Fujii and Namba, 2017*; *Mentes et al., 2018*; *Vahokoski et al., 2020*; *von der Ecken et al., 2016*), which we do not observe for myosin-V.

Notably, our data show no significant change of the helical symmetry parameters upon myosin binding, neither in rigor nor in any other state of myosin (*Table 5*). This is in stark contrast to an earlier medium-resolution study of myosin-V, which reported additional twisting of PHD-stabilized F-actin dependent upon the nucleotide state of myosin (*Wulf et al., 2016*).

It was reported that myosin-V is sensitive to the nucleotide state of F-actin and prefers young PHD-stabilized F-actin over aged F-actin-PHD (*Zimmermann et al., 2015*). We have recently shown that young ATP/ADP-Pᵢ-bound and aged ADP-bound F-actin primarily differ in their conformation of the D-loop-C-terminus interface and that actin-binding proteins like coronin-IB (*Cai et al., 2007*) probably recognize the nucleotide state of F-actin from this interface (*Merino et al., 2018*). We have furthermore shown that the short-lived ATP/ADP-Pᵢ-bound state of F-actin can be specifically stabilized using

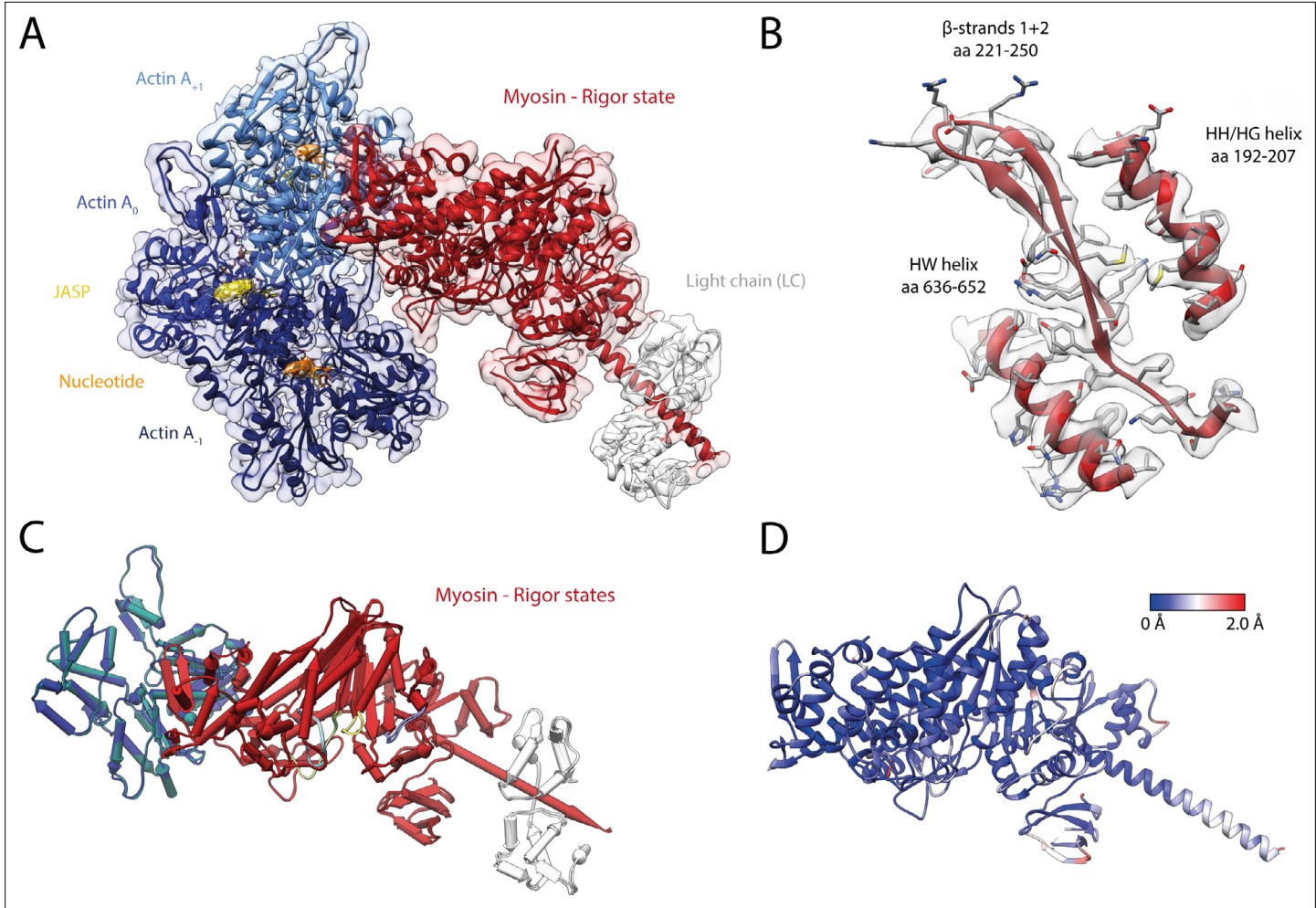

**Figure 7.** Structure of the young actomyosin-V complex in the rigor state. (**A**) Atomic model and LAFTER density map of the central myosin-V-LC subunit (red, LC: white) bound to young F-actin-JASP (shades of blue, three subunits shown, A_{-1} to A_{+1}). Nucleotides and JASP are highlighted in orange and yellow, respectively; also see *Figure 8—video 1F–H*. (**B**) Illustration of the model-map agreement within a central section of myosin. Most side chains are resolved by the post-refined density map (transparent gray). (**C**) Superposition and (**D**) color-coded root mean square deviation (RMSD) of the young and aged actomyosin-V complex in the rigor state illustrating their structural identity. Residues with increased RMSD solely localize to regions of lower local resolution and can therefore be explained by modeling inaccuracies. See *Figure 7—figure supplement 1* and Table 6 for an overview of the cryo-EM data and refinement and model building statistics, respectively. The structure of young F-actin-JASP in the absence of myosin is shown in *Figure 7—figure supplement 2*.

The online version of this article includes the following figure supplement(s) for figure 7:

**Figure supplement 1.** Overview of the cryo-EM data and resolution of young F-actin-JASP alone and in complex with myosin-V in the rigor state.

**Figure supplement 2.** Structure of young JASP-stabilized F-actin.

either PHD (*Lynen and Wieland, 1938*) or jasplakinolide (JASP) (*Crews et al., 1986*; *Pospich et al., 2020*). To reveal the structural mechanism by which myosin-V senses the nucleotide state of F-actin, we have solved the structure of myosin-V in the rigor state in complex with young JASP-stabilized F-actin (F-actin-JASP) to 3.2 Å (referred to as young actomyosin-V, *Figure 7*, *Table 6*, *Figure 7—figure supplement 1*, *Figure 1—figure supplement 1*, *Table 1*). The atomic model of myosin in this structure superimposes perfectly with the one bound to aged F-actin-PHD (*Figure 7C and D*), indicating that the nucleotide state of F-actin has no structural effect on myosin-V in the rigor state. Surprisingly, and despite having ADP-$P_i$ bound to its active site (*Figure 1—figure supplement 3*), F-actin adopts the closed D-loop state, which is characteristic for aged ADP-bound F-actin (*Figure 7*; *Merino et al., 2018*). However, a control structure of F-actin-JASP alone (3.1 Å, *Figure 7—figure supplement 1*, *Table 1*, *Table 6*, *Figure 1—figure supplement 1*) confirms that actin was successfully stabilized in

**Table 6.** Statistics of young actomyosin in the rigor state.
Refinement and model building statistics of young F-actin-JASP alone and in complex with myosin-V in the rigor state.

| | Young F-actin-JASP | | Rigor state: young F-actin-JASP + myosin-Va-LC | | | |
|---|---|---|---|---|---|---|
| | Actin only 3er/2er | Central 3er/2er | Central 1er (subtracted) | Class 1 | Class 2 | Class 4 |
| **3D refinement statistics** | | | | | | |
| Number of helical segments | 212,660 | 414,148 | 414,148 | 110,797 | 107,022 | 107,174 |
| Resolution (Å) | 3.1 | 3.2 | 3.2 | 3.6 | 3.5 | 3.6 |
| Map sharpening factor (Å$^2$) | −56 | −83 | −50 | −55 | −49 | −54 |
| **Atomic model statistics** | | | | | | |
| Non-hydrogen atoms | 8940 | 23,278 | 10,149 | 10,169 | 10,169 | 10,156 |
| Cross-correlation masked | 0.81 | 0.84 | 0.83 | 0.84 | 0.83 | 0.83 |
| MolProbity score | 1.27 | 1.29 | 1.15 | 1.24 | 1.26 | 1.23 |
| Clashscore | 5.11 | 5.46 | 3.62 | 4.66 | 4.91 | 4.57 |
| EMRinger score* | 3.11/3.08 | 2.92/2.66 | 3.11/2.92 | 2.89/2.96 | 2.99/3.39 | 2.88/2.55 |
| Bond RMSD (Å) | 0.004 | 0.004 | 0.009 | 0.005 | 0.003 | 0.004 |
| Angle RMSD (°) | 0.915 | 0.780 | 0.950 | 0.836 | 0.807 | 0.835 |
| Rotamer outliers (%) | 0.00 | 0.00 | 0.00 | 0.00 | 0.00 | 0.00 |
| Ramachandran favored (%) | 100.00 | 99.86 | 99.84 | 99.84 | 99.84 | 99.84 |
| Ramachandran outliers (%) | 0.00 | 0.00 | 0.00 | 0.00 | 0.00 | 0.00 |
| CaBLAM outliers (%) | 0.27 | 0.75 | 0.90 | 0.81 | 0.65 | 0.49 |

*Values correspond to score against the post-refined map used for real-space refinement/a map filtered to local resolution.

the desired young state, having a characteristic open D-loop conformation (*Figure 7—figure supplement 2*) and ADP-P$_i$ bound to its active site (*Figure 1—figure supplement 3*). Thus, we conclude that binding of myosin-V to young F-actin-JASP induces structural changes that ultimately result in the closed D-loop conformation (*Figure 8*, *Figure 8—video 1*, *Figure 8—figure supplement 1*), thereby abolishing the effect of JASP (*Pospich et al., 2020*). Interestingly, our data show that the open D-loop state would not clash with bound myosin (*Figure 8C and D*). The closed conformation may therefore be selected for its superior shape complementarity to myosin, which possibly establishes a strong binding interface between the D-loop and HLH motif and by doing so contributes to the high-binding affinity of the rigor state (*Figure 8*).

Our structure does not provide a structural explanation for the reported nucleotide-sensitivity of myosin-V (*Zimmermann et al., 2015*). This could be due to three, possibly complementary, reasons. First, myosin-V might be sensitive to the nucleotide state of F-actin only in certain structural states, such as the initially binding PPS (*Wulf et al., 2016*) and P$_i$R states (*Llinas et al., 2015*). Second, the structural plasticity of young ATP/ADP-P$_i$-bound F-actin (*Kueh and Mitchison, 2009*), rather than the open D-loop conformation, might be beneficial for myosin binding. Third, the open D-loop conformation might promote the formation of initial contacts with myosin-V. Once these are established, the subsequent transition from a weak- to a strong binding state potentially causes a structural transition of F-actin, eventually locking it in the closed D-loop conformation. In line with these theories, a number of biochemical and biophysical studies suggested that a structural rearrangement of F-actin and its structural plasticity are critical for proper myosin activity (*Anson et al., 1995*; *Drummond et al., 1990*; *Kim et al., 2002*; *Nishikawa et al., 2002*; *Noguchi et al., 2012*; *Oztug Durer et al., 2011*; *Prochniewicz and Thomas, 2001*; *Prochniewicz et al., 2010*). Moreover, the D-loop C-terminus interface was predicted to contribute to the initial binding interface of myosin (*Gurel et al., 2017*; *Lehman et al., 2013*; *Risi et al., 2017*; *Robert-Paganin et al., 2020*).

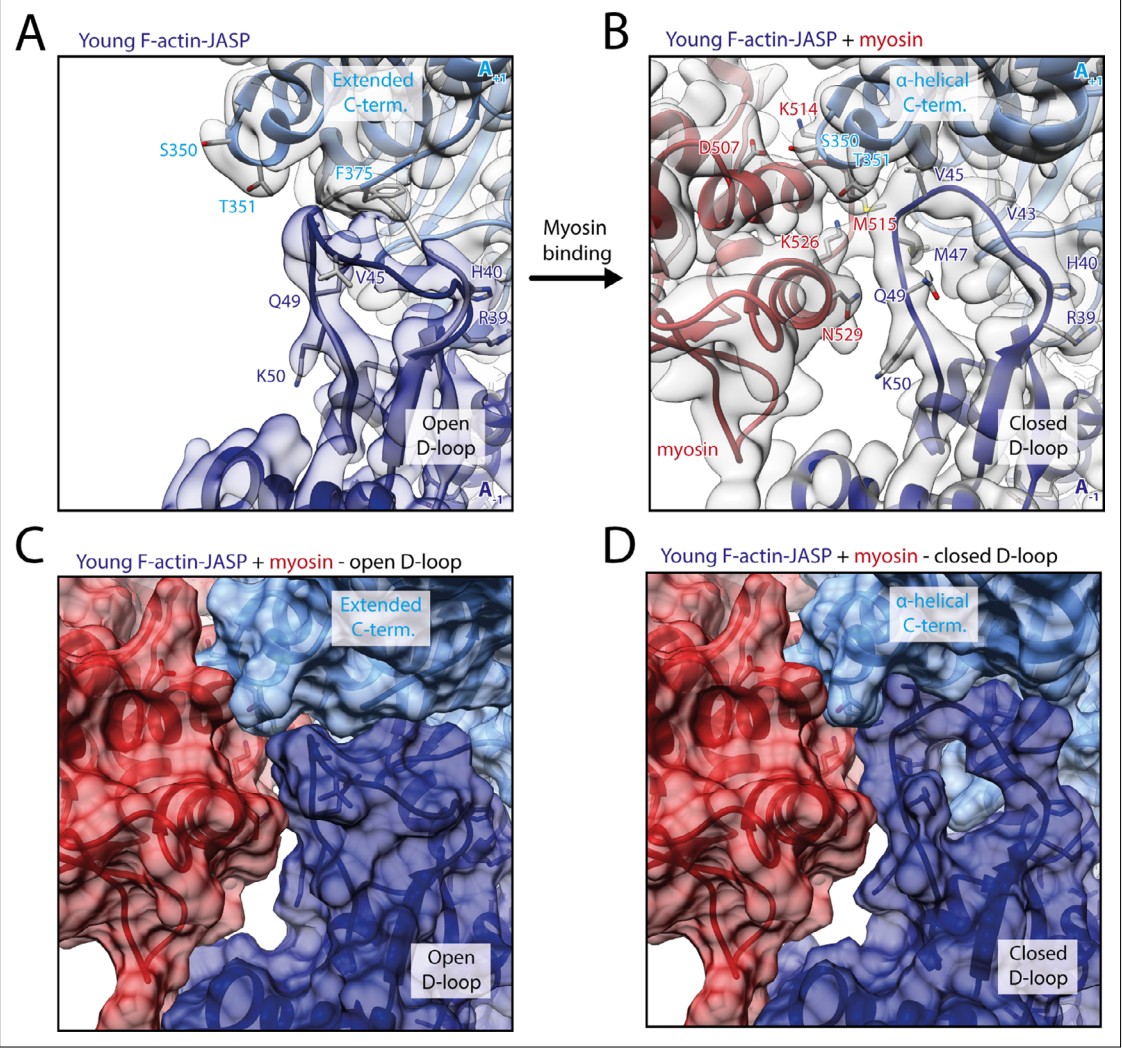

**Figure 8.** Myosin-V binding causes closure of the D-loop in young JASP-stabilized F-actin. (**A**) Atomic model and LAFTER density map of young F-actin-JASP (shades of blue, subunits $A_{-1}$ and $A_{+1}$). Before myosin binding, the D-loop primarily adopts the open conformation and the C-terminus is extended. A superimposed atomic model (gray) highlights a minor density potentially corresponding to the closed D-loop conformation. (**B**) Binding of myosin-V in the rigor state (red) causes a structural transition to the closed D-loop conformation, which comes with an α-helical C-terminus; also see *Figure 8—video 1* and *Figure 8—figure supplement 1*. (**C**) Surface representation of young F-actin-JASP (open D-loop, as shown in **A**) illustrating that the open D-loop conformation would not clash with myosin (computationally docked). (**D**) Surface representation of the young JASP-stabilized actomyosin complex (closed D-loop, as shown in **B**). See *Figure 8—figure supplement 2* for an illustration how pyrene labeling might interfere with myosin binding.

The online version of this article includes the following video and figure supplement(s) for figure 8:

**Figure supplement 1.** Myosin-V selects a specific conformation of F-actin.

**Figure supplement 2.** Pyrene labeling potentially impedes selection of the closed D-loop.

**Figure 8—video 1.** Structural changes of JASP-stabilized F-actin upon binding of myosin-V.

https://elifesciences.org/articles/73724/figures#fig8video1

Finally, the conformational selection mechanism of myosin-V offers a structural explanation for the quenching of pyrene fluorescence upon myosin binding. Pyrene conjugated to cysteine 374 in the C-terminus of F-actin has been often used to report not only actin kinetics, but also myosin binding (*Kouyama and Mihashi, 1981*). Closure of the actin-binding cleft of myosin is thought to expose pyrene to the solvent and thus cause fluorescence quenching (*Chou and Pollard, 2020*), but the exact timing and the structural basis are not yet known (*Llinas et al., 2015*; *Robert-Paganin et al., 2020*). A recent cryo-EM structure of pyrene-labeled F-actin has revealed that pyrene wedges itself between

the tip of the D-loop and the hydrophobic groove surrounding it, partially pushing the D-loop out of its binding pocket (*Chou and Pollard, 2020*). This likely interferes with myosin selecting the closed D-loop state (*Figure 8—figure supplement 2*). We furthermore suggest that myosin quenches the fluorescence of pyrene by pushing it out of its binding pocket when selecting the closed D-loop state during its transition to a strong binding state.

## Pronounced structural heterogeneity of myosin-V

To identify a potential mixture of structural states, we performed 3D classifications of signal-subtracted particles for all our data sets (*Figure 1—figure supplement 1*). Interestingly, the results indicate a continuous conformational heterogeneity of myosin-V as opposed to a mixture of several discrete structural states (see Materials and methods for details). Based on the identified 3D classes, we solved and modeled a total of 18 high-resolution (<3.7 Å) structures of actomyo-sin-V (*Figure 1—figure supplement 1*, *Table 2*, *Table 3*, *Table 4* and *Table 6*). A superposition of all structures from one data set illustrates pronounced structural flexibility of all domains, but the L50 domain, F-actin, and the actomyosin interface (*Figure 9*, also see *Figure 5*). Primarily, the U50 domain pivots and moves toward or away from the actin interface, resulting in twisting and shifting of the central transducer β-sheet, which is coupled to rotations of the N-terminal and the converter domain (*Figure 9A*). In this way, pivoting of the U50 domain leads to different lever arm positions within the 3D classes of a single data set (*Figure 9A*, *Figure 9—video 1*, *Figure 9—video 2*, *Figure 9—video 3*). The extent (~9–12°) of the relative lever arm swings is intriguing (*Figure 9A*, *Figure 9—figure supplement 1*), considering that the swing associated with $Mg^{2+}$-ADP-release is only ~9° for myosin-V (*Figure 3*).

Our data show that the conformational heterogeneity of myosin-V is not caused by variations of the active site or mixed nucleotide states (*Figure 9*). Nevertheless, the presence of a nucleotide does affect the extent of flexibility as ADP and AppNHp lead to a greater change in lever arm position (*Figure 9A*). This tendency is also reflected by the size of the respective conformational spaces when mapping all models belonging to one data set onto their principal components (PCs) using principal component analysis (PCA) (*Figure 9B*).

To impartially compare the conformations of the different nucleotide states of myosin-V, we performed a PCA of all models (*Figure 10*). The structural similarity and differences of the atomic models are well reflected by their localization within the PC space as well as their corresponding conformational spaces (*Figure 10A*). Notably, the significantly larger conformational space of the AppNHp data indicates a considerable difference to the rigor state, supporting our proposal of a PRT state. The fact that the conformational spaces of the strong-ADP and rigor state do not overlap is anticipated, given that we have oversaturated myosin with $Mg^{2+}$-ADP (see Materials and methods).

The conformational changes mapped on each PC are readily illustrated by their corresponding trajectories as well as the extreme structures along each PC (*Figure 10C*, *Figure 10—video 1*). The motions along the first and second PCs correspond to an almost perpendicular pivoting of the U50 domain, causing a twist and shift of the central transducer β-sheet and ultimately rotations of the N-terminal and converter domain. The third PC maps a rotation of the N-terminal and converter domain around the transducer, which acts as a hinge region. Since all average structures localize close to the origin of PC 3 (*Figure 10A*, *Figure 10—video 1E and F*), we suggest that this PC accounts for an inherent flexibility of the transducer β-sheet.

The rearrangements, especially along the first PC, are reminiscent of the structural transition of myosin-V upon $Mg^{2+}$-ADP release (*Figures 3 and 10C*, *Figure 10—video 1*). In line with this, we find the strong-ADP and rigor average structures to be arranged diagonally within the PC 1–PC 2 space (*Figure 10A*). This indicates that the conformational heterogeneity of myosin-V as well as the isomerization associated with $Mg^{2+}$-ADP release relies on the same principal coupling mechanism. Furthermore, this suggests that the structural transition of myosin-V along its motor cycle is driven, at least in part, by its conformational flexibility. Based on this, we therefore propose that the active site of myosin-V is not mechanically, and thus rigidly, coupled to the surrounding domains, particularly the lever arm, as previously proposed (*Fischer et al., 2005*). Rather, its coupling seems to be statistical in nature, ultimately leading to a thermodynamic ensemble of conformations within each state. The associated structural flexibility of myosin-V possibly initiates transitions between structural states by giving rise to short-lived intermediate conformations with favorable nucleotide-binding affinities. Interactions with

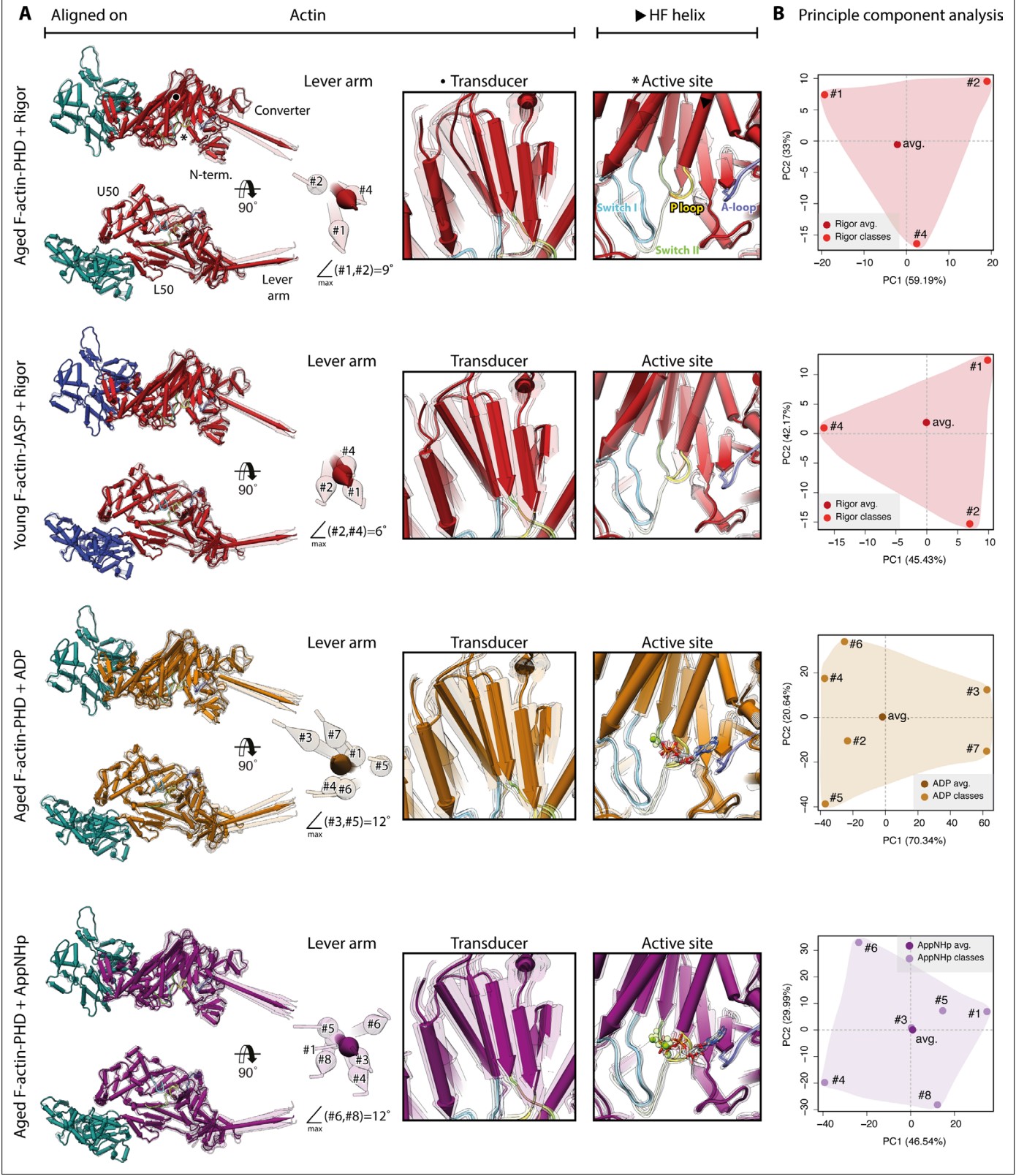

**Figure 9.** Conformational heterogeneity of myosin-V. Illustration of the conformational heterogeneity of myosin-V in the rigor (red), strong-ADP (orange), and AppNHp-bound post-rigor transition (PRT) state (purple) when bound to F-actin (aged F-actin-PHD: sea green; young F-actin-JASP: blue). (**A**) Superposition of all atomic models (central 1er, average: opaque; 3D classes: transparent) built for each state. Models were either aligned on the F-actin subunit or the HF helix (indicated by black arrowhead). Pivoting of the U50 domain in combination with shifting and twisting of the central

*Figure 9 continued on next page*

*Figure 9 continued*

transducer β-sheet results in a rotation of the N-terminal and converter domain, giving rise to a two-dimensional distribution of lever arm orientations. The extent of these changes depends on the nucleotide state and is largest in the strong-ADP and PRT state. Insets show either the transducer β-sheet (black dot) or the active site (asterisk), which basically remains unchanged within all models of one state. (**B**) Mapping of atomic models (average and 3D classes) into the first two principal components of a principal component analysis (PCA) illustrating the overall conformational space covered. Classes are labeled by their number (#1–#8; also see *Figure 1—figure supplement 1*). For a comparison of conformational extremes, see *Figure 9—figure supplement 1*. Morphs of extremes and trajectories along the principal components are visualized in *Figure 9—video 1*, *Figure 9—video 2*, and *Figure 9—video 3*. See *Figure 1—figure supplement 6* for an overview of the domain architecture of myosin.

The online version of this article includes the following video and figure supplement(s) for figure 9:

**Figure supplement 1.** Extreme conformations of myosin-V.

**Figure 9—video 1.** Structural heterogeneity of myosin-V in the strong-ADP state.

https://elifesciences.org/articles/73724/figures#fig9video1

**Figure 9—video 2.** Structural heterogeneity of myosin-V in the rigor state.

https://elifesciences.org/articles/73724/figures#fig9video2

**Figure 9—video 3.** Structural heterogeneity of myosin-V in the post-rigor transition (PRT) state (AppNHp).

https://elifesciences.org/articles/73724/figures#fig9video3

a nucleotide would consequently not trigger the transition, but merely stabilize myosin in its transient conformation, thereby promoting the transition to a new structural ensemble state.

A non-rigid, stochastic coupling of the active site of myosin-V is in good agreement with the release of $Mg^{2+}$-ADP due to an isomerization as well as the existence of the PRT state. It also provides a good explanation for the different binding affinities of the rigor and strong-ADP state. Specifically, we propose that the extent of conformational heterogeneity tunes the binding affinity rather than changes in the actomyosin interface since these are almost the same in all three nucleotide states studied (*Figure 5*). Restrictions of the conformational space by external forces, that is, load on the lever arm, could account for the load dependence of transitions within the cycle, such as the delay of ADP release under load (*Mentes et al., 2018*).

The conformational flexibility we observe (*Figures 9 and 10*) as well as our conclusions on its role in the motor cycle are in line with more than two decades of molecular spectroscopy experiments, which have primarily, but not exclusively, studied myosin-II. In particular, site-directed labeling has demonstrated that myosin is highly dynamic and that multiple, functionally relevant structural states coexist within a single biochemical state (*Forkey et al., 2003*; *Nesmelov et al., 2008*; *Nesmelov et al., 2011*; *Thomas et al., 2009*). Moreover, it was shown that neither the active site is tightly coupled to the structural domains of the motor nor are the domains themselves (*Klein et al., 2008*; *Korman et al., 2006*; *Naber et al., 2010*; *Sun et al., 2006*). Our results extend the spectroscopic data, which have already elucidated conformational amplitudes and kinetics, by directly visualizing the dynamics of myosin as well as the underlying molecular coupling.

While the agreement of our results with the spectroscopic data on myosin-II (*Thomas et al., 2009*) already suggests that statistical coupling and conformational flexibility are general features of the myosin superfamily, rather than a hallmark of myosin-V, there are additional independent indications. On the one hand, statistical coupling of the active site has also been proposed for myosin-VI based on a recovery stroke intermediate crystal structure, showing that the lever arm can partially re-prime while the active site remains unchanged (*Blanc et al., 2018*). On the other hand, conformational heterogeneity has also been reported for myosin-IE and -IB based on either crystal structures or cryo-EM data of the actomyosin complex (*Behrmann et al., 2012*; *Kollmar et al., 2002*; *Mentes et al., 2018*). Notably, a flexibility reminiscent of the one observed for myosin-V (*Figures 9 and 10*) was reported for myosin-IE in the rigor state (*Behrmann et al., 2012*). Conversely, no flexibility was described for myosin-IB, which adopts a single state in the absence of a nucleotide and two discrete states when bound to $Mg^{2+}$-ADP (*Mentes et al., 2018*). Whether these results reflect properties of specific myosins or rather current limitations of data analysis methods, for example, number of particles, low signal-to-noise ratio, robustness of 3D classifications (*Pospich and Raunser, 2018*), remains to be investigated. In general, there is little structural data on the conformational dynamics of myosin as most structures originate either from small cryo-EM data sets, which have an insufficient number of particles for extensive 3D classifications, or from X-ray crystallography. We therefore believe that the

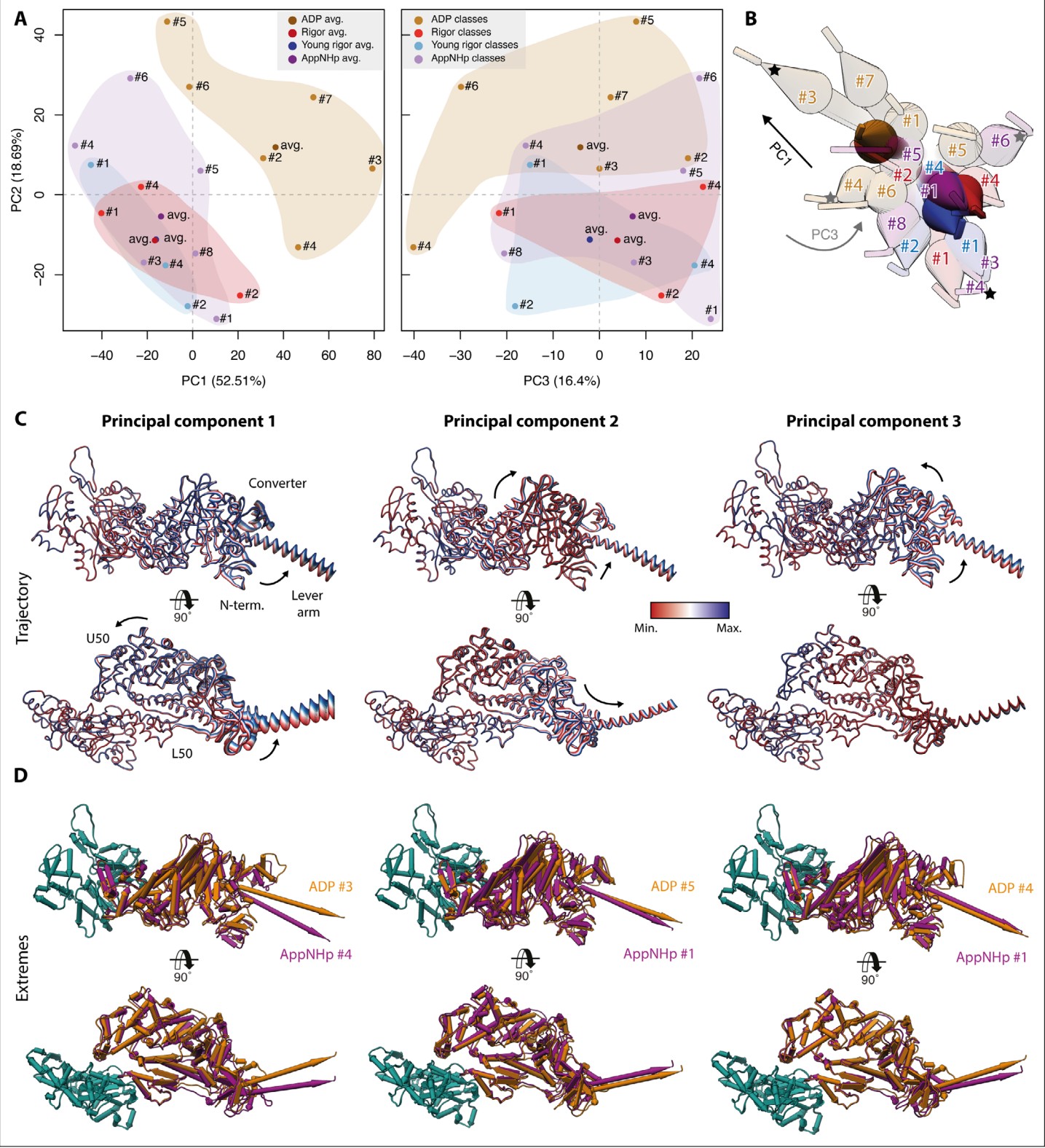

**Figure 10.** Principal component analysis of all myosin-V models. Principal component analysis of all atomic models of the actomyosin-V complex, including average and 3D class average models of the strong-ADP, rigor, and post-rigor transition (PRT) state (central actomyosin subunit only). (**A**) Mapping of atomic models into the first and second as well as the second and third principal components. Data points are colored by the state of the actomyosin-V complex (aged rigor: red; aged strong-ADP: orange; aged AppNHp-bound PRT: purple; and young rigor: blue). Atomic models of average structures are shown as opaque, and models of 3D classes as transparent. The conformational space covered within each state is indicated

*Figure 10 continued on next page*

*Figure 10 continued*

by a correspondingly colored 2D polygon. (**B**) Superposition of all lever arm positions reflecting the relative mapping of individual conformational spaces. Changes along the first and third principal components are highlighted by black and gray arrows, respectively (extremes marked with asterisks). (**C**) Color-coded trajectories along the first, second, and third principal components (red minimum, blue maximum). Arrows indicate the mapped conformational changes. (**D**) Same views as in (**C**) but showing the extreme structures along each principal component; see *Figure 9* for color code. For an animation of trajectories and morphs of the extreme structures, see *Figure 10—video 1*; and see *Figure 1—figure supplement 6* for an overview of the domain architecture of myosin.

The online version of this article includes the following video for figure 10:

**Figure 10—video 1.** Structural heterogeneity of myosin-V: principal component analysis (PCA) of all atomic models.
https://elifesciences.org/articles/73724/figures#fig10video1

structural characterization of myosin's dynamic landscape will provide novel insights into the details of force generation.

## Summary

The presented high-resolution cryo-EM structures of the actomyosin-V complex in three nucleotide states—nucleotide-free, $Mg^{2+}$-ADP, and $Mg^{2+}$-AppNHp (*Table 1*)—provide valuable insights into the structural basis of force generation. First, a comparison of the strong-ADP (*Figure 1*) and rigor state (*Figure 2*) has revealed the structural transition of myosin-V upon $Mg^{2+}$-ADP-release (*Figure 3*), which is reminiscent of the one of myosin-IB (*Mentes et al., 2018*) and yet differs in its details. Second, the structure of $Mg^{2+}$-AppNHp-bound myosin-V has uncovered a previously unseen post-rigor transition (PRT) state (*Figure 4*), which is strongly bound to F-actin and adopts a conformation resembling the rigor state. Because of the weak binding to the active site, AppNHp, and probably ATP, does not directly trigger the detachment from F-actin and thus the transition to the post-rigor state. Instead, strong nucleotide binding likely needs to be established to eventually initiate detachment.

Interestingly, and despite the differences in the F-actin-binding affinity, we find that the actin-binding interface is basically indistinguishable in all three nucleotide states (*Figure 5*), suggesting that strongly bound states utilize a common binding scheme. Furthermore, a comparison of the interface with the one of other myosins has revealed specific features of the myosin-V interface and indicates that a strong Milligan contact (*Milligan et al., 1990*) is characteristic of myosins with long lifetimes of actin-bound states and high binding affinities for ADP and F-actin, as found in high duty-ratio myosins and myosin-IB (*Laakso et al., 2008*; *Lewis et al., 2006*).

In contrast to previous reports (*Wulf et al., 2016*), our results elucidate that myosin-V hardly alters the structure of aged F-actin-PHD (*Figure 6*). Conversely, it has a remarkable effect on the structure of young F-actin-JASP, specifically selecting the closed D-loop state (*Figures 7 and 8*) and thereby overriding the 'rejuvenating effect' of JASP (*Merino et al., 2018*; *Pospich et al., 2020*). Whilst this result does not reveal the structural basis of myosin-V's nucleotide sensitivity (*Zimmermann et al., 2015*), it offers an explanation for pyrene fluorescence quenching upon myosin binding (*Kouyama and Mihashi, 1981*).

Additional heterogeneity analysis of our data revealed a pronounced structural flexibility of myosin-V (*Figures 9 and 10*), indicating a non-rigid, stochastic coupling of the active site. While the extent of flexibility is altered by the presence of a nucleotide, structural transitions of myosin-V are likely not initiated by binding of a specific nucleotide, but rather by thermodynamic fluctuations, as previously suggested for myosin-VI (*Blanc et al., 2018*).

Taken together, we have elucidated many, previously unknown details of the force generation mechanism. The general validity of these results, that is, if they are limited to myosin-V or hold for the complete myosin superfamily, as well as the possible implications of our findings has to be thoroughly tested in future studies. Structural data on how actin activates myosin and how myosin eventually detaches will surely be of interest (*Robert-Paganin et al., 2020*; *Schröder, 2020*; *Sweeney et al., 2020*). Yet, great insights could also come from the structural characterization of myosin's dynamic landscape. Finally, unraveling the structural basis of nucleotide sensitivity (*Zimmermann et al., 2015*) will further promote our understanding of the regulation of both myosin and the actin cytoskeleton (*Merino et al., 2020*).

# Materials and methods

## Key resources table

| Reagent type (species) or resource | Designation | Source or reference | Identifiers | Additional information |
|---|---|---|---|---|
| Gene (*Gallus gallus*) | MYO5A | *De La Cruz et al., 1999* | Uniprot ID:Q02440 | Unconventional myosin-Va |
| Gene (*Homo sapiens*) | MYL6B (MLC1SA) | *De La Cruz et al., 1999* | Uniprot ID:P14649 | Myosin light chain 6B/myosin LC 1 – slow-twitch muscle A isoform |
| Cell line (*Spodoptera frugiperda*) | SF9 cells | *De La Cruz et al., 1999* | | Insect cells, for baculovirus expression |
| Biological sample (*Oryctolagus cuniculus*) | Rabbit skeletal muscle acetone powder | Gift from W. Linke and A. Unger (Ruhr-Universität Bochum, Germany) | N/A | For purification of α-actin (Uniprot ID:P68135) |
| Recombinant DNA reagent | pVL1392 pVL1393 (plasmids) | *De La Cruz et al., 1999* | Invitrogen, V1392-20 | |
| Chemical compound, drug | Phalloidin (PHD) *Amanita phalloides* | Sigma-Aldrich | P2141 | For stabilization of aged ADP-bound F-actin |
| Chemical compound, drug | Jasplakinolide (JASP) | Sigma-Aldrich | J4580 | For stabilization of young ADP-$P_i$-bound F-actin |
| Chemical compound, drug | AppNHp (AMPPNP) | Jena Bioscience | NU-407-10 | |
| Chemical compound, drug | ADP | Sigma-Aldrich | A2754 | |
| Software, algorithm | TranSPHIRE | *Stabrin et al., 2020*; PMID:33177513 | v1.4–1.5.7 | |
| Software, algorithm | MotionCor2 | *Zheng et al., 2017*; PMID:28250466 | v1.1.0; v1.3.0; v1.2.6 | Within TranSPHIRE |
| Software, algorithm | GCTF | *Zhang, 2016*; PMID:26592709 | v1.06 | Within TranSPHIRE |
| Software, algorithm | crYOLO | *Wagner et al., 2020*; PMID:32627734 | v1.2.2; v1.2.4; v1.4.1 | Within TranSPHIRE |
| Software, algorithm | GPU-ISAC | *Stabrin et al., 2020*; PMID:33177513 | v1.2 and earlier | Within TranSPHIRE |
| Software, algorithm | Cinderella | *Stabrin et al., 2020*; PMID:33177513 | v0.3.1 | Within TranSPHIRE |
| Software, algorithm | SPHIRE | *Moriya et al., 2017*; PMID:28570515 | v1.3 | Helical processing pipeline, including CTF refinement and signal subtraction |
| Software, algorithm | Relion | *Scheres, 2012*; PMID:23000701 | v3.0.4 | For particle polishing and 3D classifications |
| Software, algorithm | UCSF Chimera | *Pettersen et al., 2004*; PMID:15264254 | v1.15 | |
| Software, algorithm | UCSF ChimeraX | *Goddard et al., 2018*; PMID:28710774 | v0.91 | For model building with ISOLDE |
| Software, algorithm | ISOLDE | *Croll, 2018*; PMID:29872003 | v1.0b4 | |
| Software, algorithm | Coot | *Emsley et al., 2010*; PMID:20383002 | v0.8.9.2 | |
| Software, algorithm | Phenix | *Adams et al., 2011*; *Afonine et al., 2018*; PMID:18094468 | v1.17.1 | |
| Software, algorithm | elBOW | *Moriarty et al., 2009*; PMID:19770504 | v1.17.1 | Within Phenix |
| Software, algorithm | MolProbity | *Chen et al., 2010*; PMID:20057044 | v1.17.1 | Within Phenix |
| Software, algorithm | EMRinger | *Barad et al., 2015*; PMID:26280328 | v1.17.1 | Within Phenix |
| Software, algorithm | LAFTER | *Ramlaul et al., 2019*; PMID:30502495 | v1.1 | |
| Software, algorithm | Bio3d | *Grant et al., 2006*; PMID:32734663 | v2.3-4 | Library for PCA in R |
| Software, algorithm | DynDom | *Hayward and Lee, 2002*; PMID:12463636; http://dyndom.cmp.uea.ac.uk | | Accessed October 2020 |

*Continued on next page*

*Continued*

| Reagent type (species) or resource | Designation | Source or reference | Identifiers | Additional information |
|---|---|---|---|---|
| Software, algorithm | PDBsum | *Laskowski et al., 2018*; PMID:28875543; https://www.ebi.ac.uk/pdbsum/ | | Accessed November 2020 |
| Other | Cryo-EM grids | Quantifoil (QF) | R2/1 300 mesh | |

## Protein expression and purification

Actin was purified from rabbit skeletal muscle acetone powder by cycles of polymerization and depolymerization as described previously (*Merino et al., 2018*; *Pardee and Spudich, 1982*; *Pospich et al., 2020*). Purified G-actin was flash-frozen and stored in G-actin buffer (5 mM Tris pH 7.5, 1 mM DTT, 0.2 mM $CaCl_2$, 2 mM $NaN_3$, and 0.5 mM ATP) at –80°C.

Myosin V was expressed using the baculovirus/SF9 cell expression system. To create the recombinant virus used for expression, the cDNA coding for chicken myosin-Va was truncated after the codon corresponding to Arg792. This construct encompassed the motor domain and the first light chain/calmodulin-binding site of myosin-Va. A 'Flag' tag DNA sequence (encoding GDYKDDDDK) (*Hopp et al., 1988*) was appended to the truncated myosin-V coding sequence to facilitate purification. A truncated cDNA for the LC1-sa light chain (*De La Cruz et al., 2000*) was coexpressed with the truncated myosin-V heavy chain in SF9 cells as described in *De La Cruz et al., 1999*. The cells were grown for 72 hr in medium containing 0.2 mg/ml biotin, harvested and lysed by sonication in 10 mM imidazole, pH 7.4, 0.2 M NaCl, 1 mM EGTA, 5 mM $MgCl_2$, 7% (w/v) sucrose, 2 mM DTT, 0.5 mM 4-(2-aminoethyl)benzenesuflonyl fluoride, 5 µg/ml leupeptin, and 2 mM MgATP. An additional 2 mM MgATP was added prior to a clarifying spin at 200,000 × *g* for 40 min. The supernatant was purified using FLAG-affinity chromatography (Sigma). The column was washed with 10 mM imidazole pH 7.4, 0.2 M NaCl, and 1 mM EGTA, and the myosin eluted from the column using the same buffer plus 0.1 mg/ml FLAG peptide. The fractions containing myosin were pooled and concentrated using an Amicon centrifugal filter device (Millipore) and dialyzed overnight against F-actin buffer (10 mM HEPES pH 7,5, 100 mM KCl, 2 mM $MgCl_2$, 1 mM DTT, and 1 mM $NaN_3$). Purified myosin-V-LC was flash-frozen and stored at –80°C.

## Sample preparation for cryo-EM

Aliquots of G-actin were freshly thawed and cleared by ultracentrifugation (Beckmann Rotors, TLA 120.1, 100.000 × *g*, 1 hr, 4°C). The concentration of G-actin was measured by absorption spectroscopy (Spectrophotometer DS-11, DeNovix, $E_{290\ nm} \approx 22,000\ M^{-1}\ cm^{-1}$ at 290 nm; *Hertzog and Carlier, 2005*). Polymerization was induced by adding 100 mM KCl, 2 mM $MgCl_2$, and 0.5 mM ATP. In case of young JASP-stabilized F-actin, actin was polymerized in the presence of a 2× molar excess of JASP (Sigma-Aldrich, freshly solved in DMSO, 1 mM stock). After 2 hr of incubation at room temperature, the sample was transferred to 4°C for further polymerization overnight. Filaments were collected by ultracentrifugation (Beckmann Rotors, TLA 120.1, 100.000 × *g*, 2 hr, 4°C) and pellets rinsed and resuspended in F-actin buffer (10 mM HEPES pH 7.5, 100 mM KCl, 2 mM $MgCl_2$, 1 mM DTT, 1 mM $NaN_3$) supplemented with 0.02 w/v% Tween 20 (to improve spreading of the sample droplet on the cryo-EM grid). No additional ADP or JASP was added. In case of aged PHD-stabilized F-actin, a 2× molar excess of PHD (Sigma-Aldrich, freshly solved in methanol, 1.25 mM stock) was added to resuspended filaments, which have aged, that is, hydrolyzed ATP and released the inorganic phosphate, during the overnight polymerization step. Filaments were stored at 4°C for a few hours before preparation of cryo-EM grids.

Aliquots of myosin-V-LC were freshly thawed, diluted 1:1 with F-actin buffer, and cleared by centrifugation (Eppendorf centrifuge 5424R, 21,000 × *g*, 5 min, 4°C). The concentration was determined by absorption spectroscopy (Spectrophotometer DS-11, DeNovix, $E_{280\ nm} \approx 106,580\ M^{-1}\ cm^{-1}$ at 280 nm).

## Cryo-EM grid preparation and screening

To avoid bundling of actomyosin filaments, F-actin was decorated with myosin-V-LC on the grid, as described previously (*von der Ecken et al., 2016*). A freshly glow-discharged holey-carbon grid (QF R2/1 300 mesh, Quantifoil) was mounted to a Vitrobot cryoplunger (Thermo Fisher). 3 µl of F-actin

(3–4 µM) were applied onto the front of the grid and incubated for 60 s. Excess solution was manually blotted from the side using blotting paper (Whatman No. 4). Immediately, 3 µl of myosin-V-LC (3–13 µM) were applied onto the grid and incubated for 30 s. The grid was automatically blotted for 9 s (blot force –15 or –25, drain time 0–1 s) and plunged into liquid ethane. The temperature was set to 13°C for all samples but the AppNHp sample, where either 4 or 25°C were used (two settings and data sets, see *Table 1*).

Myosin was kept in F-actin buffer and was only diluted and supplemented with a nucleotide and Tween 20 immediately before application to the grid to avoid any adverse effects. When preparing the strong-ADP state, myosin was diluted 1:1 in a 2× ADP buffer (F-actin buffer with 40 mM $MgCl_2$, 4 mM ADP, and 0.04 w/v% Tween 20). For the rigor samples, myosin was diluted in F-actin buffer supplemented with 0.02 w/v% Tween 20. AppNHp-bound samples were prepared in analogy to rigor samples, but additional 5 mM AppNHp and 4 mM $MgCl_2$ were added. As AppNHp hydrolyzes spontaneously, only freshly solved (10 mM HEPES pH 8.0, 1 mM DTT, 1 mM $NaN_3$, and 2 mM $MgCl_2$) or recently frozen AppNHp was used. Ion-pair reversed-phase chromatography experiments using freshly solved AppNHp indicated a purity of ≥98%, with 1.5% $AppNH_2$ (hydrolysis product) and no preferential binding of $AppNH_2$ to myosin. Thus, $AppNH_2$ does not get enriched in the active site of myosin-V as it is the case for F-actin (*Cooke and Murdoch, 1973*). To increase the binding affinity of AppNHp-bound myosin to F-actin (*Konrad and Goody, 2005*), the concentration of potassium chloride in the myosin sample buffer was reduced to 10–13 mM KCl by dilution with F-actin buffer without KCl. F-actin samples were diluted using F-actin buffer supplemented with 0.02 w/v% Tween 20. After dilution to the final concentration, the PHD-stabilized F-actin samples contained 0.4–0.9% methanol.

Protein concentrations were adjusted empirically based on the overall concentration on the grid and decoration of actin filaments. The concentration of myosin required to saturate F-actin (3–4 µM) strongly depended on the nucleotide state; while 3–4 µM myosin were sufficient in case of the rigor and strong-ADP state, 10–13 µM myosin were required for the AppNHp sample, even though the salt concentration of the buffer was lowered to increase the binding affinity.

Grids were screened on a Talos Arctica microscope (Thermo Fisher) operated at 200 kV and equipped with a Falcon III direct detector (Thermo Fisher).

In total, six different samples were plunged, screened, and imaged; also see *Table 1*. On the one hand, aged PHD-stabilized F-actin was decorated with myosin-V-LC in three different nucleotide states, that is, in the absence of a nucleotide and bound to either $Mg^{2+}$-ADP or $Mg^{2+}$-AppNHp (aged rigor, ADP, and AppNHp). For the AppNHp-bound sample, two data sets were collected from grids that were plunged using different incubation temperatures, that is, 4°C or 25°C. On the other hand, young JASP-stabilized F-actin was imaged on its own and in complex with myosin-V-LC in the rigor state (young F-actin and rigor). The corresponding grids were prepared in one plunging session, that is, within a short time frame of 1–2 hr, using the same JASP-stabilized F-actin sample.

## Cryo-EM data acquisition

Data sets were acquired on Titan Krios microscopes (FEI Thermo Fisher) operated at 300 kV and equipped with a X-FEG using EPU. Specifically, data sets of the rigor and strong-ADP state were acquired on a standard Krios (Cs 2.7 mm, pixel size 1.06 Å), while a Cs-corrected Krios (pixel size 1.10 Å) was used for the remaining data sets. Equally dosed frames were collected using a K2 Summit (super-resolution mode, Gatan) direct electron detector in combination with a GIF quantum-energy filter (Bioquantum, Gatan) set to a slit width of 20 eV. For every hole, four micrographs consisting of 40 frames were collected close to the carbon edge, resulting in a total electron dose of ~79–82 e$Å^{-2}$ within an exposure time of 15 s. The defocus was varied within a range of ~0.4–3.2 µm. Acquisition details of all six data sets (aged rigor, ADP, and AppNHp 4°C + 25°C as well as young F-actin and rigor) including pixel size, electron dose, defocus range, and the total number of images collected are summarized in *Table 1*. Data acquisitions were monitored and evaluated live using TranSPHIRE (*Stabrin et al., 2020*).

## Cryo-EM data processing

Data sets were automatically preprocessed on-the-fly during the data acquisition using TranSPHIRE (*Stabrin et al., 2020*). Preprocessing included drift correction and dose weighting by MotionCor2 (*Zheng et al., 2017*), CTF estimation using GCTF (*Zhang, 2016*), and particle picking with crYOLO

(*Wagner et al., 2020*; *Wagner et al., 2019*) (filament mode, box distance 26–27 px equivalent to one rise of ~27.5 Å, minimum number of boxes 6) for all data sets. The latest version of TranSPHIRE, which was used for the processing of the AppNHp data sets, also supported automatic, on-the-fly particle extraction (box size 320 px, filament width 200 px) as well as batch-wise 2D classification (batch size 13k, filament width 200 px, radius 150 px, 60–100 particles per class), 2D class selection, and 3D refinement using software of the SPHIRE package (*Moriya et al., 2017*). In particular, a GPU-accelerated version of ISAC (*Stabrin et al., 2020*; *Yang et al., 2012*) and the deep-learning 2D class selection tool Cinderella (*Wagner, 2020*) were used. For all other data sets, particles were extracted and 2D classified after data collection using analogous settings and helical SPHIRE 1.3 (*Moriya et al., 2017*; *Stabrin et al., 2020*). Particles that were not accounted during the initial, batch-wise 2D classification, for example, because they represent rare views, were merged and inputted to another round of 2D classification until no more stable classes were found. All micrographs were assessed manually and images sorted based on ice and protein quality, resulting in a removal of 6–36% of the data sets; see *Table 1* for details. Particles contributing to classes found 'good' by either Cinderella or manual inspection and belonging to micrographs of good quality were written to virtual particle stacks for further processing in 3D.

As an initial 3D refinement and 3D classification revealed no differences in the overall structure of myosin in the two AppNHp data sets, plunged at 4°C and 25°C, corresponding particles were merged for further processing. The final number of particles ranged from 212,660 (young JASP-stabilized F-actin) to 2,446,218 (combined AppNHp data sets); see *Table 2*, *Table 3*, *Table 4* and *Table 6* for details. A concise overview of all key processing steps including the number of particles and nominal resolutions can be found in *Figure 1—figure supplement 1*.

All data sets were processed using the helical refinement program sp_meridien_alpha.py implemented in SPHIRE 1.3 (*Moriya et al., 2017*; *Stabrin et al., 2020*). In contrast to other helical refinement routines, SPHIRE does not refine or apply any helical symmetry, and thereby avoids possible symmetrization pitfalls. Instead, the software offers the usage of constraints tailored to helical specimen, for example, on the tilt angle and shift along the filament, to guide the refinement (also see Methods section of *Pospich et al., 2021*). For all 3D refinements, the tilt angle was softly restrained to the equator during exhaustive searches (`--theta_min 90 --theta_max 90 --howmany 10`). The shift along the filament axis was furthermore limited to plus or minus half of the rise (`--helical_rise 27.5`) to avoid shifts larger than one subunit. Finally, the smear (number of views considered for the reprojection of each particle) was reduced to a combined weight of 90% (`--ccfpercentage 90`). An initial 3D reference was created from the atomic model of a previously published actomyosin complex in the rigor state (PDB:5JLH, without tropomyosin; *von der Ecken et al., 2016*) and filtered to 25 Å using EMAN2 (*Tang et al., 2007*) and SPHIRE (*Moriya et al., 2017*). For the initial 3D refinement, a sampling angle of 3.7°, filament width of 120 px and a radius of 144 px (45% of the box size), but no 3D mask, was used. Based on the resulting 3D density map, a wide mask covering the central 85% of the filament was created. This map and mask were then used to run a fresh, global 3D refinement using the same settings as before. Based on the results of this refinement, particles were CTF refined within SPHIRE (*Moriya et al., 2017*) providing the nominal resolution according to the $FSC_{0.143}$-criterion. CTF-refined particles were locally 3D refined using the final map of the previous 3D refinement filtered to 4 Å as reference. The fine angular sampling typically used in local refinements makes helical restraints superfluous as projections parameters can only locally relax anyways. For this reason, particles were locally refined using the non-helical 3D refinement program sp_meridien.py in combination with a sampling angle of 0.9°, a shift range of 2 px, and a shift step size of 0.5 px. In case of the young F-actin and young rigor data sets, the resolution could be further improved by particle polishing in Relion 3.0.4 (*Scheres, 2012*; *Zivanov et al., 2018*). For this purpose, refinement results were converted to Relion star format using sp_sphire2relion.py. Metadata of the initial motion correction step required for polishing were automatically created by TranSPHIRE and were directly provided. Polished particles were transferred back to SPHIRE and passed through another round of local 3D refinement using the same settings as before.

To focus the refinement on the central part of the filament, a wide mask containing the central three actin and central two myosin-V-LC subunits including all ligands (subvolume referred to as central 3er/2er map) was created and applied in a subsequent local 3D refinement. Post refinement of the resulting half maps using a central 3er/2er mask yielded maps with average resolutions ranging from

2.9 to 3.2 Å according to the $FSC_{0.143}$-criterion; see *Figure 1—figure supplement 1*, *Table 2*, *Table 3*, *Table 4* and *Table 6* for details.

With the aim to further improve the density of myosin, the signal of all subunits but the central actomyosin subunit (subvolume referred to as central 1er map) was subtracted from the 2D particle images within SPHIRE 1.3 (*Moriya et al., 2017*). Particles were additionally recentered to bring the center of mass close to the center of the box. Signal-subtracted particles were subjected to another round of local 3D refinement applying a central 1er mask and filtering the centered reference map to 3.5 Å. Although post refinement of the resulting half maps using a central 1er mask did not yield density maps of higher nominal resolution, the map quality of especially myosin could be significantly improved; see *Figure 1—figure supplements 1–2*, *Table 2*, *Table 3*, *Table 4*, *Table 6*, and *Figure 7—figure supplement 2* for details.

The anisotropic quality of the final central 1er maps suggested structural heterogeneity within myosin. For this reason, signal-subtracted particles and corresponding projection parameters were transferred to and 3D classified in Relion 3.0.4 (*Scheres, 2012*). As domain movement was assumed to be small and to reduce the risk of overrefinement, 3D alignment was deactivated (`--skip_align`) and the resolution strictly limited to 8 Å (`--strict_highres_exp 8`). The final central 1er map filtered to 15 Å was inputted as a reference, while a corresponding wide mask was applied and solvent flattening and CTF correction activated. The regularization parameter T and number of classes K were empirically adjusted. While a parameter of T = 40 (`--tau2_fudge 40`) proved well suited for all data sets, finding a suitable number of classes posed a challenge. Running multiple 3D classifications with different numbers of classes resulted in classes of various, related structural states with little overlap, that is, classes of different runs could not be matched as they did generally not superimpose. The same was true when rerunning a 3D classification job using the same settings but a different seed. These results suggest a continuous structural heterogeneity of myosin in contrast to several discrete states. While software tailored to the characterization of cryo-EM data exhibiting continuous structural states has recently been published (*Zhong et al., 2021*), it proved unsuitable for the processing of signal-subtracted actomyosin filaments due to the need of 3D masking. To characterize the structural heterogeneity of myosin-V by standard 3D classification in Relion 3.0.4 (*Scheres, 2012*) as good as possible, the number of 3D classes was optimized experimentally to yield the highest number of classes with a resolution and map quality sufficient for atomic modeling (≤3.7 Å). To do so, multiple 3D classifications with varying number of classes, for example, from 2 to 12, were performed and particles split into subsets according to the classification results. Subsets were then transferred to SPHIRE and individually subjected to a local 3D refinement from stack (no reference required, same settings as before). Eventually, each subset was post-refined and the resulting map manually assessed. In the end, the 3D classification that yielded the most maps of high quality was chosen. In this way, a total of 18 high-resolution maps (referred to as 3D class averages or 3D classes) were achieved for the four actomyosin data sets. Corresponding subsets contained 81,757 to 365,722 particles; see *Table 2*, *Table 3*, *Table 4* and *Table 6* for details. An overview of all refined maps, associated resolutions, and the underlying number of particles is given in *Figure 1—figure supplement 1*.

To ease the interpretation of maps as well as model building, all final maps, that is, central 3er/2er, central 1er, and 3D class averages, were additionally filtered to local resolution using SPHIRE 1.3 (*Moriya et al., 2017*) and denoised using LAFTER (*Ramlaul et al., 2019*).

## Model building, refinement, and validation

Previous cryo-EM structures of PHD-stabilized aged F-actin (PDB: 6T20; *Pospich et al., 2020*) and JASP-stabilized young F-actin (PDB: 5OOD; *Merino et al., 2018*) were used as starting models for F-actin in the rigor actomyosin complexes (aged and young rigor). The models of PHD and JASP were replaced by single-residue initial models generated from SMILES strings by elBOW (*Moriarty et al., 2009*) within Phenix (*Adams et al., 2011*) using the `--amber` option. The corresponding cif constraints libraries were used for all further refinements. A rigor-like crystal structure of the myosin-V-LC complex (PDB: 1OE9; *Coureux et al., 2003*) was used as an initial model for myosin and the bound light chain within the aged rigor structure. Stubs were replaced by full residues, and residues that are missing in the crystal structure, but are resolved in the cryo-EM density map, were added manually in Coot (*Debreczeni and Emsley, 2012*; *Emsley et al., 2010*). For all other models, that is, of the ADP, AppNHp, and young rigor state, the final refined model of the PHD-stabilized rigor

actomyosin complex was used as a starting model. Initial models of nucleotides (ADP and AppNHp) are based on previous cryo-EM and crystal structures of myosin (PDB: 6C1D; *Mentes et al., 2018*; and PDB: 1MMN; *Gulick et al., 1997*). Starting models were rigid-body fitted into the density map using UCSF Chimera (*Pettersen et al., 2004*) and ligands were coarsely refined in Coot (*Debreczeni and Emsley, 2012*; *Emsley et al., 2010*) prior to model building.

Atomic models of the central actomyosin subunit, consisting of one F-actin, myosin, LC, and PHD/JASP molecule (central 1er), were refined using ISOLDE (*Croll, 2018*) within UCSF ChimeraX (*Goddard et al., 2018*). For this purpose, hydrogens were added to the starting model using the addh command in UCSF Chimera (*Pettersen et al., 2004*) and manually adjusted when necessary. Custom residue definitions for PHD and JASP were created based on the elBOW output within the ISOLDE shell. To reliably model both high- and medium-resolution features, several maps, for example, filtered to nominal or local resolution and sharpened by different B-factors, were loaded to ISOLDE. Maps filtered by LAFTER (*Ramlaul et al., 2019*) were also loaded for visual guidance, but excluded from the refinement (weight set to 0, MDFF deactivated). All density maps were segmented based on the starting model using the color zone tool within UCSF Chimera (*Pettersen et al., 2004*) to exclude density not corresponding to the central actomyosin subunit.

Each refinement in ISOLDE was started with a 2–3 min all atom simulation to reduce the overall energy of the system. Afterward, overlapping stretches of the protein and atoms within close vicinity were successively adjusted and refined. When necessary, rotamer and secondary structure restraints were introduced. After passing through the complete protein complex once, the quality of the model was assessed using the metrics provided by ISOLDE, that is, Ramachandran plot, rotamer outlier, and clash score, and outliers were locally addressed. Residues not resolved by the electron density map, for example, due to flexibility, were not included in the respective atomic model, while incompletely resolved side chains were set to most likely rotamers.

The density corresponding to the light chain was of insufficient quality for reliable model building. Hence, the model of the light chain was kept fixed during refinements in ISOLDE. Afterward, the reference crystal structure (PDB: 1OE9; *Coureux et al., 2003*) was rotamer-optimized in Coot and rigid-body fitted into the density using UCSF Chimera.

Finally, atomic models were real-space refined in Phenix (*Adams et al., 2011*; *Afonine et al., 2018*) against a sharpened density map filtered to nominal resolution ($FSC_{0.143}$). To only relax and validate the model but prohibit large changes, local grid search, rotamer, and Ramachandran restraints were deactivated and the starting model was used as a reference. Furthermore, NCS and secondary structure restraints were applied and cif libraries provided for PHD and JASP.

Only models of the central actomyosin subunit (central 1er) were built in ISOLDE. Atomic models of subsets, that is, 3D class averages, were built starting from the average, all-particle model, and the corresponding ISOLDE/UCSF ChimeraX session including restraints. Whereas average models of different states, that is, rigor, ADP, and AppNHp, were built within new sessions to avoid any bias. Atomic models consisting of three actin and two myosin-LC subunits (central 3er/2er) were assembled from the models of the monomeric complex (central 1er) by rigid-body fitting in UCSF Chimera. The filament interface was manually inspected in Coot and side chain orientations adjusted when necessary. Finally, the multimeric model was real-space refinement in Phenix.

After real-space refinement, the residue assignment of PHD was changed from a single non-standard residue to a hepta-peptide consisting of TRP-EEP-ALA-DTH-CYS-HYP-ALA. All atomic models were assessed and validated using model-map agreement (FSC, CC), MolProbity (*Chen et al., 2010*), and EMRinger (*Barad et al., 2015*) statistics.

In total, 27 atomic models were built based on density maps with a resolution ranging from 2.9 Å to 3.7 Å; models include 4 central 1er and 5 central 3er/2er all-particle models as well as 18 models representing subsets identified by 3D classification (see *Table 2*, *Table 3*, *Table 4* and *Table 6* for details).

## Structural analysis and visualization

Figures and movies were created with UCSF Chimera (*Pettersen et al., 2004*) and modified using image or movie processing software when required.

For the visualization of myosin and the actomyosin interface, central 1er (central actomyosin subunit) and central 3er/2er (central three F-actin and two myosin molecules) models and maps

are shown, respectively, as they include all important contact sites and are best resolved. Models protonated by H++ (*Anandakrishnan et al., 2012*) at pH 7.5 with HIC replaced by HIS were used for all surface representations. To optimally visualize features of different local resolution, a variety of maps are displayed within figures and movies (also see legends). Specifically, LAFTER maps are used to visualize the complete actomyosin structure and features of lower resolution, while post-refined maps are shown in close-up views, for example, of the active site.

Relative rotation angles of the lever arm were computed as angles between axes created for the corresponding helices in Chimera (*Pettersen et al., 2004*) using default settings.

Protein-protein and protein-ligand interactions were analyzed with PDBsum (*Laskowski, 2009*). Conformational changes and structural heterogeneity of the central 1er models were characterized by PCA using the Bio3d library (*Grant et al., 2006*) in R (*R Core Team, 2017*). Initially, model sequences were aligned using the *pdbaln* method. With the help of the methods *core.find* and *pdbfit*, models were then superimposed on an automatically determined structural stable core, which encompasses almost the complete F-actin subunit and parts of the HLH-motif in the L50 domain. PCA was performed running *px.xray*, excluding gaps within the sequence and ligands. Data points were manually grouped and colored based on the underlying data set and type of model, that is, average model vs. 3D class average. For the direct visualization of PCA results, trajectories along each principal component were exported using *mktrj.pca* and morphed in UCSF Chimera (*Pettersen et al., 2004*). Mobile domains within myosin (central 1er, chain A) and their motion were identified and analyzed using DynDom (*Hayward and Lee, 2002*).

## Data availability

The atomic models and cryo-EM maps are available in the PDB (*Burley et al., 2019*) and EMDB databases (*Lawson et al., 2011*) under the following accession numbers: aged PHD-stabilized actomyosin-V in the strong-ADP state: 7PM5, EMD-13521 (central 1er), 7PM6, EMD-13522 (central 3er/2er), 7PM7, EMD-13523 (class 2), 7PM8, EMD-13524 (class 3), 7PM9, EMD-13525 (class 4), 7PMA, EMD-13526 (class 5), 7PMB, EMD-13527 (class 6), 7PMC, EMD-13528 (class 7); aged PHD-stabilized actomyosin-V in the rigor state: 7PLT, EMD-13501 (central 1er), 7PLU, EMD-13502 (central 3er/2er), 7PLV, EMD-13503 (class 1), 7PLW, EMD-13504 (class 3) and 7PLX, EMD-13505 (class 4); aged PHD-stabilized actomyosin-V in the PRT state: 7PMD, EMD-13529 (central 1er), 7PME, EMD-13530 (central 3er/2er), 7PMF, EMD-13531 (class 1), 7PMG, EMD-13532 (class 3), 7PMH, EMD-13533 (class 4), 7PMI, EMD-13535 (class 5), 7PMJ, EMD-13536 (class 6), 7PML, EMD-13538 (class 8); young JASP-stabilized actomyosin-V in the rigor state: 7PLY, EMD-13506 (central 1er), 7PLZ, EMD-13507 (central 3er/2er), 7PM0, EMD-13508 (class 1), 7PM1, EMD-13509 (class 2), 7PM2, EMD-13510 (class 4); and young JASP-stabilized F-actin: 7PM3, EMD-13511. The data sets generated during the current study are available from the corresponding author upon reasonable request.

## Acknowledgements

We thank E Sirikia for the purification of myosin-V from SF9 cells. We thank O Hofnagel and D Prumbaum for assistance with data collection. We thank W Linke and A Unger for providing us with muscle acetone powder. We thank RS Goody for his friendly advice. This work was supported by the Max Planck Society (to SR), the European Research Council under the European Union's Horizon 2020 Programme (ERC-2019-SyG, grant 856118) (to SR), the Centre national de la recherche scientifique (to AH), the Agence nationale de la recherche (grant ANR-17-CE11-0029-01) (to AH), and the National Institutes of Health (grant R01-DC009100) (to HLS). SP was supported as a fellow of Studienstiftung des deutschen Volkes.

## Additional information

### Funding

| Funder | Grant reference number | Author |
|---|---|---|
| Max-Planck-Gesellschaft | | Stefan Raunser |
| Horizon 2020 | ERC-2019-SyG 856118 | Stefan Raunser |
| Agence Nationale de la Recherche | ANR-17-CE11-0029-01 | Anne Houdusse |
| National Institutes of Health | R01-DC009100 | H Lee Sweeney |
| Centre National de la Recherche Scientifique | | Anne Houdusse |
| Studienstiftung des Deutschen Volkes | | Sabrina Pospich |

The funders had no role in study design, data collection and interpretation, or the decision to submit the work for publication.

### Author contributions

Sabrina Pospich, Data curation, Formal analysis, Investigation, Methodology, Validation, Visualization, Writing - original draft, Writing - review and editing; H Lee Sweeney, Anne Houdusse, Conceptualization, Writing - review and editing; Stefan Raunser, Conceptualization, Funding acquisition, Project administration, Supervision, Writing - review and editing

### Author ORCIDs

Sabrina Pospich (ID) http://orcid.org/0000-0002-5119-3039
H Lee Sweeney (ID) http://orcid.org/0000-0002-6290-8853
Anne Houdusse (ID) http://orcid.org/0000-0002-8566-0336
Stefan Raunser (ID) http://orcid.org/0000-0001-9373-3016

### Decision letter and Author response

Decision letter https://doi.org/10.7554/eLife.73724.sa1
Author response https://doi.org/10.7554/eLife.73724.sa2

## Additional files

### Supplementary files

• Transparent reporting form

### Data availability

The atomic models and cryo-EM maps are available in the PDB (Burley et al., 2018) and EMDB databases (Lawson et al., 2011), under following accession numbers: aged PHD-stabilized actomyosin-V in the strong-ADP state: 7PM5, EMD-13521 (central 1er), 7PM6, EMD-13522 (central 3er/2er), 7PM7, EMD-13523 (class 2), 7PM8, EMD-13524 (class 3), 7PM9, EMD-13525 (class 4), 7PMA, EMD-13526 (class 5), 7PMB, EMD-13527 (class 6), 7PMC, EMD-13528 (class 7) ; aged PHD-stabilized actomyosin-V in the rigor state: 7PLT, EMD-13501 (central 1er), 7PLU, EMD-13502 (central 3er/2er), 7PLV, EMD-13503 (class 1), 7PLW, EMD-13504 (class 3) and 7PLX, EMD-13505 (class 4); aged PHD-stabilized actomyosin-V in the PRT state: 7PMD, EMD-13529 (central 1er), 7PME, EMD-13530 (central 3er/2er), 7PMF, EMD-13531 (class 1), 7PMG, EMD-13532 (class 3), 7PMH, EMD-13533 (class 4), 7PMI, EMD-13535 (class 5), 7PMJ, EMD-13536 (class 6), 7PML, EMD-13538 (class 8); young JASP-stabilized actomyosin-V in the rigor state: 7PLY, EMD-13506 (central 1er), 7PLZ, EMD-13507 (central 3er/2er), 7PM0, EMD-13508 (class 1), 7PM1, EMD-13509 (class 2), 7PM2, EMD-13510 (class 4); and young JASP-stabilized F-actin: 7PM3, EMD-13511.

The following datasets were generated:

| Author(s) | Year | Dataset title | Dataset URL | Database and Identifier |
|---|---|---|---|---|
| Pospich S, Sweeney H L, Houdusse A, Raunser S | 2021 | Cryo-EM structure of the actomyosin-V complex in the strong-ADP state (central 1er) | https://www.rcsb.org/structure/7PM5 | RCSB Protein Data Bank, 7PM5 |
| Pospich S, Sweeney H L, Houdusse A, Raunser S | 2021 | Cryo-EM structure of the actomyosin-V complex in the strong-ADP state (central 3er/2er) | https://www.rcsb.org/structure/7PM6 | RCSB Protein Data Bank, 7PM6 |
| Pospich S, Sweeney H L, Houdusse A, Raunser S | 2021 | Cryo-EM structure of the actomyosin-V complex in the strong-ADP state (central 1er, class 2) | https://www.rcsb.org/structure/7PM7 | RCSB Protein Data Bank, 7PM7 |
| Pospich S, Sweeney H L, Houdusse A, Raunser S | 2021 | Cryo-EM structure of the actomyosin-V complex in the strong-ADP state (central 1er, class 3) | https://www.rcsb.org/structure/7PM8 | RCSB Protein Data Bank, 7PM8 |
| Pospich S, Sweeney H L, Houdusse A, Raunser S | 2021 | Cryo-EM structure of the actomyosin-V complex in the strong-ADP state (central 1er, class 4) | https://www.rcsb.org/structure/7PM9 | RCSB Protein Data Bank, 7PM9 |
| Pospich S, Sweeney H L, Houdusse A, Raunser S | 2021 | Cryo-EM structure of the actomyosin-V complex in the strong-ADP state (central 1er, class 5) | https://www.rcsb.org/structure/7PMA | RCSB Protein Data Bank, 7PMA |
| Pospich S, Sweeney H L, Houdusse A, Raunser S | 2021 | Cryo-EM structure of the actomyosin-V complex in the strong-ADP state (central 1er, class 6) | https://www.rcsb.org/structure/7PMB | RCSB Protein Data Bank, 7PMB |
| Pospich S, Sweeney H L, Houdusse A, Raunser S | 2021 | Cryo-EM structure of the actomyosin-V complex in the strong-ADP state (central 1er, class 7) | https://www.rcsb.org/structure/7PMC | RCSB Protein Data Bank, 7PMC |
| Pospich S, Sweeney H L, Houdusse A, Raunser S | 2021 | Cryo-EM structure of the actomyosin-V complex in the rigor state (central 1er) | https://www.rcsb.org/structure/7PLT | RCSB Protein Data Bank, 7PLT |
| Pospich S, Sweeney H L, Houdusse A, Raunser S | 2021 | Cryo-EM structure of the actomyosin-V complex in the rigor state (central 3er/2er) | https://www.rcsb.org/structure/7PLU | RCSB Protein Data Bank, 7PLU |
| Pospich S, Sweeney H L, Houdusse A, Raunser S | 2021 | Cryo-EM structure of the actomyosin-V complex in the rigor state (central 1er, class 1) | https://www.rcsb.org/structure/7PLV | RCSB Protein Data Bank, 7PLV |
| Pospich S, Sweeney H L, Houdusse A, Raunser S | 2021 | Cryo-EM structure of the actomyosin-V complex in the rigor state (central 1er, class 2) | https://www.rcsb.org/structure/7PLW | RCSB Protein Data Bank, 7PLW |
| Pospich S, Sweeney H L, Houdusse A, Raunser S | 2021 | Cryo-EM structure of the actomyosin-V complex in the rigor state (central 1er, class 4) | https://www.rcsb.org/structure/7PLX | RCSB Protein Data Bank, 7PLX |
| Pospich S, Sweeney H L, Houdusse A, Raunser S | 2021 | Cryo-EM structure of the actomyosin-V complex in the post-rigor transition state (AppNHp, central 1er) | https://www.rcsb.org/structure/7PMD | RCSB Protein Data Bank, 7PMD |

*Continued on next page*

*Continued*

| Author(s) | Year | Dataset title | Dataset URL | Database and Identifier |
|---|---|---|---|---|
| Pospich S, Sweeney H L, Houdusse A, Raunser S | 2021 | Cryo-EM structure of the actomyosin-V complex in the post-rigor transition state (AppNHp, central 3er/2er) | https://www.rcsb.org/structure/7PME | RCSB Protein Data Bank, 7PME |
| Pospich S, Sweeney H L, Houdusse A, Raunser S | 2021 | Cryo-EM structure of the actomyosin-V complex in the post-rigor transition state (AppNHp, central 1er, class 1) | https://www.rcsb.org/structure/7PMF | RCSB Protein Data Bank, 7PMF |
| Pospich S, Sweeney H L, Houdusse A, Raunser S | 2021 | Cryo-EM structure of the actomyosin-V complex in the post-rigor transition state (AppNHp, central 1er, class 3) | https://www.rcsb.org/structure/7PMG | RCSB Protein Data Bank, 7PMG |
| Pospich S, Sweeney H L, Houdusse A, Raunser S | 2021 | Cryo-EM structure of the actomyosin-V complex in the post-rigor transition state (AppNHp, central 1er, class 4) | https://www.rcsb.org/structure/7PMH | RCSB Protein Data Bank, 7PMH |
| Pospich S, Sweeney H L, Houdusse A, Raunser S | 2021 | Cryo-EM structure of the actomyosin-V complex in the post-rigor transition state (AppNHp, central 1er, class 5) | https://www.rcsb.org/structure/7PMI | RCSB Protein Data Bank, 7PMI |
| Pospich S, Sweeney H L, Houdusse A, Raunser S | 2021 | Cryo-EM structure of the actomyosin-V complex in the post-rigor transition state (AppNHp, central 1er, class 6) | https://www.rcsb.org/structure/7PMJ | RCSB Protein Data Bank, 7PMJ |
| Pospich S, Sweeney H L, Houdusse A, Raunser S | 2021 | Cryo-EM structure of the actomyosin-V complex in the post-rigor transition state (AppNHp, central 1er, class 8) | https://www.rcsb.org/structure/7PML | RCSB Protein Data Bank, 7PML |
| Pospich S, Sweeney H L, Houdusse A, Raunser S | 2021 | Cryo-EM structure of the actomyosin-V complex in the rigor state (central 1er, young JASP-stabilized F-actin) | https://www.rcsb.org/structure/7PLY | RCSB Protein Data Bank, 7PLY |
| Pospich S, Sweeney H L, Houdusse A, Raunser S | 2021 | Cryo-EM structure of the actomyosin-V complex in the rigor state (central 3er/2er, young JASP-stabilized F-actin) | https://www.rcsb.org/structure/7PLZ | RCSB Protein Data Bank, 7PLZ |
| Pospich S, Sweeney H L, Houdusse A, Raunser S | 2021 | Cryo-EM structure of the actomyosin-V complex in the rigor state (central 1er, young JASP-stabilized F-actin, class 1) | https://www.rcsb.org/structure/7PM0 | RCSB Protein Data Bank, 7PM0 |
| Pospich S, Sweeney H L, Houdusse A, Raunser S | 2021 | Cryo-EM structure of the actomyosin-V complex in the rigor state (central 1er, young JASP-stabilized F-actin, class 2) | https://www.rcsb.org/structure/7PM1 | RCSB Protein Data Bank, 7PM1 |

*Continued on next page*

*Continued*

| Author(s) | Year | Dataset title | Dataset URL | Database and Identifier |
|---|---|---|---|---|
| Pospich S, Sweeney H L, Houdusse A, Raunser S | 2021 | Cryo-EM structure of the actomyosin-V complex in the rigor state (central 1er, young JASP-stabilized F-actin, class 4) | https://www.rcsb.org/structure/7PM2 | RCSB Protein Data Bank, 7PM2 |
| Pospich S, Sweeney H L, Houdusse A, Raunser S | 2021 | Cryo-EM structure of young JASP-stabilized F-actin (central 3er) | https://www.rcsb.org/structure/7PM3 | RCSB Protein Data Bank, 7PM3 |

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
