## [Editor Report]

This work obtains an atomic-level understanding of mechanochemical coupling and the structural elements that lead to mechanical and chemical diversity of the cytoskeletal motor, myosin-V.

---

## [Decision Letter]

**Decision letter after peer review:**

Thank you for submitting your article "High-resolution structures of the actomyosin-V complex in three nucleotide states provide insights into the force generation mechanism" for consideration by *eLife*. Your article has been reviewed by 3 peer reviewers, and the evaluation has been overseen by Samara Reck-Peterson as Reviewing Editor and José Faraldo-Gómez as Senior Editor. The following individual involved in review of your submission has agreed to reveal their identity: E Michael Ostap (Reviewer #3).

Essential revisions:

The reviewers generally agreed that this paper will be important to the myosin field as it provides a molecular-level understanding of mechanochemical coupling, and of the structural elements that underlie the mechanical and chemical diversity of cytoskeletal motors. Essential revisions are listed by number below. In addition, several concerns were brought up by the reviewers, which should be addressed in a revision as well.

1) The authors should acknowledge that molecular spectroscopists have long known that different biochemical states of myosins are dynamic with varying amplitudes and kinetics of flexibility.

2) There is little experimental support (biochemical experiments) to demonstrate that the AppNHp state is indeed a post-rigor transitional state. Either more work needs to be performed to show that this is the case, or this claim needs to be scaled back.

3) The statement that Pi release requires completion of the power stroke is incorrect, as the lead head of myo5, when bound to actin, can release Pi, but the converter is still in a pre-power state conformation (Walker et al., 2000, Nature). This is an important point, and this statement needs correcting.

4) Please see additional essential revisions requested by Reviewer #1.

*Reviewer #1:*

Pospich et al., have used CryoEM and image processing to generate a 3D structure of the motor domain of myosin V bound to F-actin in three different states. These include the Mg.ADP state in which myosin V is strongly bound to actin but has not yet released ADP, the nucleotide free state (also known as the rigor state) which occurs once ADP is released, and a third state in which a non-hydrolysable ATP analogue (AppNHp, also known as AMPPNP) is weakly bound in the nucleotide binding pocket, and may represent the binding of ATP to the rigor state immediately preceding myosin detachment from actin, termed the post-rigor transition state. These structures build on those of Mg.ADP myosin V and nucleotide-free myosin V bound to F-actin at lower resolution (8Å) reported in an earlier paper from Wulf et al., (PNAS, 2016). The data build on previous rigor structures from a wide range of different myosins from multiple laboratories, to which they show strong similarities, such as a closed actin binding cleft in the myosin motor domain and a post-powerstroke lever arm position.

Structure of Mg.ADP myosin V bound to actin

The authors first show the details of a new Mg.ADP myosin V state bound to actin (Figure 1 and related supplementary material), which nicely shows how ADP is co-ordinated within the nucleotide binding site, including the identification of important hydrophobic interactions between adenosine and residues 111-116, termed the A-loop.

Perhaps not surprisingly, they find that this new structure differs from a previous crystal structure of myosin V with Mg.ADP in the nucleotide binding site, which was obtained by soaking in Mg.ADP into a crystal of the rigor myosin motor domain (with a single light chain: Coureaux et al., 2004). One would expect that adding Mg.ADP to a preformed crystal is unlikely to generate large structural changes. In contrast, the cryoEM structure of Mg.ADP myosin bound to actin is more likely to resemble the cryoEM structure of Mg.ADP Myosin-IB bound to actin (Mentes et al., 2018) as the authors report, as the complex is frozen in the presence of Mg.ADP.It would be helpful to include this discussion in the text.

Structure of nucleotide free myosin V bound to actin

The authors go on to describe the nucleotide free state of myosin V bound to F-actin (Figure 2 and related supplementary material). The structure of this state is entirely consistent with previous rigor structures of myosin bound to F-actin: the actin binding cleft is closed, and the lever arm is in a post-power stroke conformation. The actin binding interface is highly similar to that in the ADP state for myosin V, but the position of the lever arm is rotated by ~9° as shown in the previous lower resolution structures of myosin V structure bound to actin in the presence and absence of ADP (Wulf et al., 2016). Not surprisingly, the structures are not identical to rigor crystal structures of myosin in the absence of actin.

Clear changes are demonstrated for the nucleotide binding site, between ADP and rigor states, as expected (Figure 3 and related video), and it is suggested that a shift in the A-loop might stabilise a resulting twist in the transducer and N-terminal domain rotation, revealing an important role for this loop in coupling of the nucleotide binding site to other regions in myosin, helping to result in the amplification of small changes in this site to large lever arm swings. As the authors discuss, similar structural transitions have been reported elsewhere.

There is then some speculation as to whether this lever arm swing, which is different is size for different myosins, might be related to the mechanosensitivity of myosin. However, it is worth reminding readers that myosin 1B is monomeric, while myosin V is dimeric. In myosin V the lead head is actually trapped in a pre-powerstroke conformation (Walker et al., 2000, Nature), with strain between the two heads 'gating' ADP release, such that its release is slower from the lead head. This is perhaps different to the mechanical load myosin 1B might experience.

Structure of AppNHp-myosin V bound to actin

The overall structure of this state (Figure 4) is remarkably similar to that of the rigor state (Figure 4 supplementary figures). The myosin is strongly bound, and the lever is in the post-powerstroke lever arm orientation. The nucleotide binding site is also similar to that found in rigor. The authors speculate that this structure represents a post-rigor transition state and compare it to the myosin V ADP crystal structure, in which ADP was soaked into the rigor crystal. The assertion that this really is a new post-rigor transition (PRT) state appears somewhat tenuous to me. The difficulty is, of course, that as soon as ATP binds strongly, the motor will dissociate from actin, and it wasn't entirely clear how this provided new insights into how myosin detaches from F-actin. There is no clear evidence that this state really is the PRT state.

Comparison of the actomyosin interface for all three states:

The authors show that the overall actomyosin-V interface is highly similar in each of the three states (Figure 5), is similar to previous structures of this interface for other myosins bound to actin and demonstrate a few myosin V specific interactions.

Importantly, it is unlikely that the actomyosin interface would change when ADP is released from myosin. If it did change, then the expectation would be that actin activates ADP release. However, work from Howard White's laboratory has demonstrated that the ADP release rate from myosin V is unchanged between experiments in which actin is either present or absent. It would be helpful to add this discussion to the paper.

Overall actin structure in unbound and rigor actomyosin states, and in young and old actin (Figures6-8):

A comparison of the structure of actin in the presence and absence of rigor myosin V demonstrates the importance of the D-loop in actin in this interaction, as found previously for other myosins. However, as expected, actin structure is mostly the same in the presence and absence of myosin V. It is interesting that the authors find no difference between the rigor actomyosin structures for old and young actin, suggesting that the rigor state of myosin is not sensitive to the nucleotide state of F-actin, and that the D-loop is found in the closed conformation, characteristic of 'old' actin. However, the speculation that myosin V specifically selects 'old' actin or could promote phosphate release from actin, and the following discussion about possible structural re-arrangements, is highly speculative and not well supported by experimental data.

Structural heterogeneity

The authors use their existing datasets to explore variability or heterogeneity in the structures (Figure 9-10 and supplementary material). It is important to note here that the numbers of particles in the AppNHp dataset is 3 times as large as that in the other two data sets (rigor and ADP). The main source of heterogeneity appears to be in the transducer and lever arm, but not in the actin binding interface, as might be expected.

There is the possibility that the flexibility or position of the lever might be linked to where it is located in the frozen sample. Thin films of ice have the potential to compress the sample, which could affect the lever, and might explain why the light chain is poorly resolved compared to the rest of the molecule. It would be interesting to know if the flexed molecules are in plane (e.g. uncompressed) or all near the top and bottom of the liquid film, or randomly distributed?

Actomyosin ATPase cycle

The authors attempt to place the three structures described here in the context of a complete actomyosin ATPase cycle, adding in previously published structures, many of which are crystal structures of myosin in the absence of actin (Figure 11 and supplementary material). This figure (and the supplementary figures) could have been generated without any of the data presented in this new paper.

Critically, what we are really missing here is the structure of the pre-power stroke state as myosin first binds to actin. This is a difficult state to capture, due to the weak binding of myosin ATP to actin, and no-one has managed to capture this state yet. It is likely to have a different structure to the PiR and PPS states presented in Figure 11, as its structure will be influence by binding to actin. It is unclear why there is a mix of myosin V and myosin VI structures in this figure, as the sequences of these two myosins are not strongly conserved, and it is unclear if the states found in myosin VI are representative of those in myosin V as shown in this pathway. Overall, it is unclear how this part of the paper adds to our overall knowledge, or how it provides us with an unprecedented insight into force generation as it does not show how myosin attaches to actin and accelerates Pi release.

Overall, this is a solid paper, with newer higher resolution structures of the rigor and ADP actomyosin V states, which will be interesting to the field, demonstrate that the actomyosin interface is essentially highly similar, and that any structural variability mostly resides in the lever.

The paper does not 'provide unprecedented insight into force generation' as claimed as we still do not know how myosin attaches to actin and what structural changes occur that accelerate of Pi release. This is still the key missing step. However, the new high-resolution structures of ADP and rigor myosin V bound to actin are a useful contribution to the field.

There is very little experimental justification (biochemical experiments) that demonstrate the AppNHp state is indeed a post-rigor transitional state. Either more work needs to be performed to show that this is the case, or this claim needs to be scaled back.

In examining heterogeneity, the data for each actomyosin state could have been divided up into more classes with a lower number in each class, which might be expected to show an increase in heterogeneity. How was the decision made on the number of classes? Did the different number of particles available for the AppNHp class affect this analysis? The authors should explain their reasoning here in the revised paper.

In Supplementary Figure 2D and E – A comparison of maps constructed from the same particle number would be a fairer and more informative comparison. This would address the question as to whether differences in the resolution of the cryoEM maps from different states are due to heterogeneity or differences in particle number.

The statement that Pi release requires completion of the power stroke is incorrect, as the lead head of myo5, when bound to actin, can release Pi, but the converter still in a pre-power state conformation (Walker et al., 2000, Nature). This is an important point, and this statement needs correcting.

Regarding the statement 'Our data indicate that the powerstroke of myosin-V amounts to ~ 77°, in contrast to a previous estimate of only 58° based on medium resolution cryo-EM data (Wulf et al., 2016). It is unclear why this angle changes when everything else seems the same as before. Is it possible that this is due to a change in the way the molecules have been super-posed before looking at global movements? Could the authors clarify this and re-analyse if appropriate.

Recommendations for improving the writing and presentation.

The paper is overlong, and there is considerable speculation in places that is not required and which detracts from the paper.

The authors should revise their discussion of the comparison of the Mg.ADP myosin V structures as indicated in the public review.

The authors should comment on the data from Howard White, that suggests we would not expect to find a change in actin structure between ADP and rigor states, as indicated in the public review.

Generally, the writing was clear, but the paper is overly long, and contains quite a bit of speculation that could easily be removed without detracting from the key findings. The authors should shorten and focus the paper. There are a few odd uses of English in places, which would be improved by careful proof reading.

I recommend removing Figure 11 and associated figures completely, as they do not add to the paper. In fact, I think they somewhat detract from the key findings of the paper concerning the two main ADP and rigor actomyosin states. Moreover, it is unclear how relevant the intermediate states from myosin VI are? They might be on pathway, but the sequence is not strongly conserved, so probably not. Suppl 1 and 2 (for Figure 11) – are mainly just re-summarising figures and are not needed.

A clearer explanation of the methods for generating the PC plots would be helpful. How was the Bio3D library used?

*Reviewer #2:*

This manuscript presents a detailed structural model of the myosin-V motor cycle by characterizing three actin-bound structural intermediates with different nucleotide states (strong-ADP, nucleotide free (rigor) and non-hydrolyzable ATP (post-rigor transition)) at near atomic resolution. These results supports and expands on details of the myosin-V mechanochemical cycle that could be inferred from previous structural work, including x-ray crystal structures and lower resolution cryo-EM studies of myosin-V. Additionally, a novel conformation is reported of ATP-analog bound, actin bound myosin-V, exhibiting striking structural similarities with the rigor state. Notably, this is the first high-resolution structure of ATP-analog bound myosin on an actin filament that has been reported. This conformation is interpreted by the authors as a post-rigor transition (PRT) state that could rationalize how ATP recognizes the rigor state before promoting detachment from F-actin. Finally, by collecting large datasets the authors were also able to generate an assessment of heterogeneity/mobility of these states which informs structural transitions between these states.

This manuscript is packed with a lot of details (and structures) but organizes them nicely in the context of existing literature, avoids overinterpretation and overall represents a solid contribution to the field. There are some concerns with two of the stated conclusions that do not seem consistent with the data presented, that need to be addressed. However, overall the manuscript is highly polished. The readability could benefit by trimming or moving to the supplementary material several main figures that are not central to the main findings.

The following points should be addressed to improve the work:

1. One conclusion from the manuscript, quoted from the summary is that "Structural transitions of myosin-V are hence likely not initiated by binding of a specific nucleotide, but rather by thermodynamic fluctuations, as previously suggested for myosin-VI". This statement doesn't seem consistent with the PCA results (Figure 10 A) showing that the ADP conformational space does not strongly overlap with that of rigor. Neither does this statement appear consistent with the authors' proposal that PRT state isomerizes after nucleotide binding to release from actin. Likewise, for page 17 where it is stated that "Our data show that the conformational heterogeneity of myosin-V is not caused by variations of the active site or mixed nucleotide states (Figure 9). Nevertheless, the presence of a nucleotide does affect the extent of flexibility, as ADP and AppNHp lead to a greater change in lever arm position (Figure 9A)." These statements seem contradictory and the authors need to clarify the ambiguity.

2. Under the "Conservation and specificity of the actomyosin-V interface" section on page 11, the last sentence says "Differences in the affinity might therefore not be linked to altered contacts, but rather to the degree of structural flexibility inherent to each state (see below)". This is an interesting statement, but the discussion below or in other sections don't address this. The authors need to expand on this.

3. The atomic model of strong ADP myosin-V is one of the more significant results since only two other myosins have been solved to near atomic resolution in this nucleotide state. It therefore seems important to provide a comparison of active site features (ie., Magnesium coordination etc) for the available ADP actomyosin co-complex structures, at least as a supplementary figure.

*Reviewer #3:*

The paper by Pospich provides important new structural insights into actomyosin-V mechanochemical coupling. The work provides new high-resolution structures of several states of myosin-V, provides new details on the relationship between myosin-binding and actin structure, and gives interesting insights into the structural heterogeneity of myosin-V.

The paper is written in a way that provides testable predictions of the importance of specific residue-residue interactions to nucleotide binding, and the conformational changes that result in lever arm tilting upon ADP release. A strength of the paper is the thoughtfulness of the supplemental videos that relate to the figures.

---

## [Author Response]

1) The authors should acknowledge that molecular spectroscopists have long known that different biochemical states of myosins are dynamic with varying amplitudes and kinetics of flexibility.

We have adjusted the manuscript accordingly and now properly introduce and reference the work of molecular spectroscopists.

2) There is little experimental support (biochemical experiments) to demonstrate that the AppNHp state is indeed a post-rigor transitional state. Either more work needs to be performed to show that this is the case, or this claim needs to be scaled back.

Biochemical experiments supporting our proposal have been published before. We have added details about these experiments to the revised manuscript to support our reasoning.

3) The statement that Pi release requires completion of the power stroke is incorrect, as the lead head of myo5, when bound to actin, can release Pi, but the converter is still in a pre-power state conformation (Walker et al., 2000, Nature). This is an important point, and this statement needs correcting.

We have adjusted the manuscript to clarify this point, including a reference to Walker et al.

4) Please see additional essential revisions requested by Reviewer #1.

See answers below.

Reviewer #1:Pospich et al., have used CryoEM and image processing to generate a 3D structure of the motor domain of myosin V bound to F-actin in three different states. These include the Mg.ADP state in which myosin V is strongly bound to actin but has not yet released ADP, the nucleotide free state (also known as the rigor state) which occurs once ADP is released, and a third state in which a non-hydrolysable ATP analogue (AppNHp, also known as AMPPNP) is weakly bound in the nucleotide binding pocket, and may represent the binding of ATP to the rigor state immediately preceding myosin detachment from actin, termed the post-rigor transition state. These structures build on those of Mg.ADP myosin V and nucleotide-free myosin V bound to F-actin at lower resolution (8Å) reported in an earlier paper from Wulf et al., (PNAS, 2016). The data build on previous rigor structures from a wide range of different myosins from multiple laboratories, to which they show strong similarities, such as a closed actin binding cleft in the myosin motor domain and a post-powerstroke lever arm position.Structure of Mg.ADP myosin V bound to actinThe authors first show the details of a new Mg.ADP myosin V state bound to actin (Figure 1 and related supplementary material), which nicely shows how ADP is co-ordinated within the nucleotide binding site, including the identification of important hydrophobic interactions between adenosine and residues 111-116, termed the A-loop.Perhaps not surprisingly, they find that this new structure differs from a previous crystal structure of myosin V with Mg.ADP in the nucleotide binding site, which was obtained by soaking in Mg.ADP into a crystal of the rigor myosin motor domain (with a single light chain: Coureaux et al., 2004). One would expect that adding Mg.ADP to a preformed crystal is unlikely to generate large structural changes. In contrast, the cryoEM structure of Mg.ADP myosin bound to actin is more likely to resemble the cryoEM structure of Mg.ADP Myosin-IB bound to actin (Mentes et al., 2018) as the authors report, as the complex is frozen in the presence of Mg.ADP.It would be helpful to include this discussion in the text.

To avoid confusion and shorten the paper, we have removed the comment in line 172-174 comparing the cryo-EM strong-ADP state with the weak-ADP rigor-like crystal structure. Based on the recommendation of reviewer #2, we have slightly extended the comparison of the strong-ADP state of different myosins (now describing how differences in the localization of switch I result in varying positions of the Mg^2+^ ion) and included a new supplementary figure (Figure 1—figure supplement 2).

Structure of nucleotide free myosin V bound to actinThe authors go on to describe the nucleotide free state of myosin V bound to F-actin (Figure 2 and related supplementary material). The structure of this state is entirely consistent with previous rigor structures of myosin bound to F-actin: the actin binding cleft is closed, and the lever arm is in a post-power stroke conformation. The actin binding interface is highly similar to that in the ADP state for myosin V, but the position of the lever arm is rotated by ~9{degree sign} as shown in the previous lower resolution structures of myosin V structure bound to actin in the presence and absence of ADP (Wulf et al., 2016). Not surprisingly, the structures are not identical to rigor crystal structures of myosin in the absence of actin.

With the aim to focus the paper, we have shortened the comparison with the rigor-like crystal structure.

Clear changes are demonstrated for the nucleotide binding site, between ADP and rigor states, as expected (Figure 3 and related video), and it is suggested that a shift in the A-loop might stabilise a resulting twist in the transducer and N-terminal domain rotation, revealing an important role for this loop in coupling of the nucleotide binding site to other regions in myosin, helping to result in the amplification of small changes in this site to large lever arm swings. As the authors discuss, similar structural transitions have been reported elsewhere.

We thank the reviewer for the precise summary of our findings. Yet, we want to emphasize that our work in general is not just a reproduction, extension or support of findings that have been reported elsewhere (as one could easily conclude from the summary). Structures of the strong-ADP and rigor state have been solved before. However, these were either limited in resolution or for other myosins, such as myosin IB and recently myosin XV. As shown by our detailed comparison of available actomyosin structures, significant differences exist within the myosin superfamily and account for the varying motor properties. The structural details that can be transferred from one myosin to the next are hence limited when it comes to understand how myosins are tuned for their diverse functions in the cell. Therefore, high-resolution structures remain critical for understanding not only the conservation and divergence of the myosin superfamily, but also the molecular details of force generation. In addition, we describe many novel aspects of myosin, including the first high-resolution structure of an AppNHp-bound myosin, myosin-V’s pronounced structural heterogeneity (which could only be characterized in detail because of the unprecedented number of high-resolution structures we generated), myosin’s effect on young JAS-stabilized F-actin as well as the role of the A-loop.

There is then some speculation as to whether this lever arm swing, which is different is size for different myosins, might be related to the mechanosensitivity of myosin. However, it is worth reminding readers that myosin 1B is monomeric, while myosin V is dimeric. In myosin V the lead head is actually trapped in a pre-powerstroke conformation (Walker et al., 2000, Nature), with strain between the two heads 'gating' ADP release, such that its release is slower from the lead head. This is perhaps different to the mechanical load myosin 1B might experience.

The point raised by the reviewer is interesting and correct. However, the state of the ADP-bound lead head under strain exerted by the rear head is not directly linked to the data presented here, since our structures correspond to structures of the (monomeric) motor domain without strain. The reference we cite with regard to the force sensitivity of myosin-V (Veigel, Schmitz, Wang, and Sellers, 2005) has also worked with a single myosin head, making their results comparable to the ones for myosin-IB. We therefore do not think that raising the point of a myosin being monomeric or dimeric would help the reader to understand our reasoning. To focus the paper and avoid speculations, we have shortened the discussion on the force-sensitivity in the revised manuscript.

Structure of AppNHp-myosin V bound to actinThe overall structure of this state (Figure 4) is remarkably similar to that of the rigor state (Figure 4 supplementary figures). The myosin is strongly bound, and the lever is in the post-powerstroke lever arm orientation. The nucleotide binding site is also similar to that found in rigor. The authors speculate that this structure represents a post-rigor transition state and compare it to the myosin V ADP crystal structure, in which ADP was soaked into the rigor crystal. The assertion that this really is a new post-rigor transition (PRT) state appears somewhat tenuous to me. The difficulty is, of course, that as soon as ATP binds strongly, the motor will dissociate from actin, and it wasn't entirely clear how this provided new insights into how myosin detaches from F-actin. There is no clear evidence that this state really is the PRT state.

Earlier kinetic data published by Yengo et al., (Biochemistry, 2002) demonstrated that the AppNHp state has a weaker affinity for F-actin than either the strong-ADP or rigor state. Specifically, De la Cruz et al., (PNAS, 1999) measured the Kd for the rigor = 4.9 10^-12^ M and ADP = 7.6 10^-9^ M states; which are significantly lower than the one of AppNHp-bound myosin-V = 0.3 10^-6^ M (Yengo et al., 2002). Thus, the AppNHp-bound myosin-V favors the dissociation from F-actin, as it is also evident from the fact that higher concentrations of myosin were required to achieve decoration of F-actin (10-13 µM instead of 3-4 µM for rigor/ADP). AppNHp is furthermore a well-established ATP-analog in the myosin field, having been used in various crystal structures before. Considering the biochemical data and the similarity of the AppNHp-bound state to the rigor state, our proposal of a post-rigor transition state comes naturally to mind.

As the biochemical data haven’t been included in our reasoning before, we have adjusted the text as follows

“Our prior kinetic studies (De La Cruz et al., 1999; Yengo et al., 2002) demonstrated that AppNHp reduces the binding affinity of myosin-V for F-actin by > 5,000 fold as compared to the rigor state, thus favoring dissociation. A weakened affinity is also supported by the higher concentrations required to achieve decoration of F-actin with myosin in the AppNHp state (see Methods). AppNHp also induces greater structural flexibility in myosin-V (see below) as compared to the rigor state, which may facilitate the transition to a detached state. Based on the presented structural and prior kinetic studies, we propose that our AppNHp-bound myosin-V structure represents a post-rigor transition (PRT) state that allows to visualize how ATP binds in the rigor state, prior to the transition that involves a switch I movement and promotes detachment of myosin from F-actin.”

Comparison of the actomyosin interface for all three states:The authors show that the overall actomyosin-V interface is highly similar in each of the three states (Figure 5), is similar to previous structures of this interface for other myosins bound to actin and demonstrate a few myosin V specific interactions.Importantly, it is unlikely that the actomyosin interface would change when ADP is released from myosin. If it did change, then the expectation would be that actin activates ADP release. However, work from Howard White's laboratory has demonstrated that the ADP release rate from myosin V is unchanged between experiments in which actin is either present or absent. It would be helpful to add this discussion to the paper.

It is unclear which study of Howard White the reviewer is quoting for demonstrating that actin does not activate ADP release, as the literature is extensive in showing the opposite, including the initial characterization of Myosin V in the De la Cruz et al., 1999 paper (which used the same myosin-V construct we used for our structures). In addition, we found an NIH grant abstract of Galkin and White, where the authors write: “Despite a detailed knowledge of the kinetics of actomyosin ATP hydrolysis in which actin accelerates the dissociation of Pi and ADP from myosin, our understanding of the molecular mechanism(s) by which the free energy of ATP hydrolysis is coupled to the production of work and movement by myosin motors remains elusive.”

It is precisely because actin does activate ADP release that we thought that the actin interface could differ between the rigor and strong-ADP state, and that this had not been seen earlier due to lack of sufficient resolution and classification of actomyosin states.

Overall actin structure in unbound and rigor actomyosin states, and in young and old actin (Figures 6-8):A comparison of the structure of actin in the presence and absence of rigor myosin V demonstrates the importance of the D-loop in actin in this interaction, as found previously for other myosins. However, as expected, actin structure is mostly the same in the presence and absence of myosin V. It is interesting that the authors find no difference between the rigor actomyosin structures for old and young actin, suggesting that the rigor state of myosin is not sensitive to the nucleotide state of F-actin, and that the D-loop is found in the closed conformation, characteristic of ‘old’ actin. However, the speculation that myosin V specifically selects ‘old’ actin or could promote phosphate release from actin, and the following discussion about possible structural re-arrangements, is highly speculative and not well supported by experimental data.

Since our data cannot explain the nucleotide sensitivity of myosin-V previously reported by Zimmermann et al., we think it is helpful to propose possible reasons for the lack of structural differences and therefore prefer to keep this discussion (prev. lines 457-474). However, we have removed the speculation about the role of myosin V on the remodeling of the actin cytoskeleton (lines 450-456) in the revised version of the manuscript.

Structural heterogeneityThe authors use their existing datasets to explore variability or heterogeneity in the structures (Figure 9-10 and supplementary material). It is important to note here that the numbers of particles in the AppNHp dataset is 3 times as large as that in the other two data sets (rigor and ADP). The main source of heterogeneity appears to be in the transducer and lever arm, but not in the actin binding interface, as might be expected.

We want to note, that the AppNHp data set is a combination of two data sets (also see methods) and that it was not our aim to generate data with equal numbers of particles. The number of particles also differs for the rigor and ADP data sets – Rigor: 300k, ADP: 870k, AppNHp: 2,446k. As we have experimentally adjusted the number of classes for each data set (see Methods and answer below), we do not see how the size of the data set should affect the significance of our results.

There is the possibility that the flexibility or position of the lever might be linked to where it is located in the frozen sample. Thin films of ice have the potential to compress the sample, which could affect the lever, and might explain why the light chain is poorly resolved compared to the rest of the molecule. It would be interesting to know if the flexed molecules are in plane (e.g. uncompressed) or all near the top and bottom of the liquid film, or randomly distributed?

We agree that the thickness of the vitreous ice layer and interactions with the air-water interface could alter the conformation of a protein and thereby cause heterogeneity. It can also result in (partial) denaturation resulting in lower resolutions. However, it is very unlikely that this is the case for our data. First, we have averaged actomyosin particles from continuous, µm long, helical filaments. Because of the helical symmetry, only a small fraction of myosins would be exposed to the air-water interface. Consequently, we would assume to find a single conformation for the majority of myosin heads (which is not the case, instead classes are approx. equally populated). Second, mapping of particles based on their conformation (different 3D classes) onto the original 2D micrographs did not reveal any correlation, but appeared randomly (see Author response image 1). Localization of particles within the ice layer, i.e. near the top and bottom as requested by the reviewer, would require tomographic data, which we have not collected. Yet, per-particle defocus values, which depend on the relative position of particles in the ice layer, do not indicate any correlation between particle position within the layer and its conformation (see Author response image 1). We are thus convinced that the observed heterogeneity is an intrinsic feature of myosin-V.

**Author response image 1. sa2fig1:** 

Actomyosin ATPase cycleThe authors attempt to place the three structures described here in the context of a complete actomyosin ATPase cycle, adding in previously published structures, many of which are crystal structures of myosin in the absence of actin (Figure 11 and supplementary material). This figure (and the supplementary figures) could have been generated without any of the data presented in this new paper.Critically, what we are really missing here is the structure of the pre-power stroke state as myosin first binds to actin. This is a difficult state to capture, due to the weak binding of myosin ATP to actin, and no-one has managed to capture this state yet. It is likely to have a different structure to the PiR and PPS states presented in Figure 11, as its structure will be influence by binding to actin. It is unclear why there is a mix of myosin V and myosin VI structures in this figure, as the sequences of these two myosins are not strongly conserved, and it is unclear if the states found in myosin VI are representative of those in myosin V as shown in this pathway. Overall, it is unclear how this part of the paper adds to our overall knowledge, or how it provides us with an unprecedented insight into force generation as it does not show how myosin attaches to actin and accelerates Pi release.

We strongly disagree with the reviewer in the points that (a) Figure 11 could have been generated without the data presented in this paper (the PRT state is completely new and no high-resolution structures were available for the strong-ADP and rigor states), and (b) that this part does not add to our overall knowledge. We have indeed not solved the initial binding state of myosin, but also never claimed that we did. Yet, there is much to learn from our structures about the molecular details of the motor cycle (i.e. transition between PPS to strong-ADP and PRT to Post-Rigor), which previously could only be speculated about, as high-resolution structures from the same myosin are required. We also clearly reasoned why we have included the myosin-VI structures (namely because these states were not solved for any other myosin) and point out that the structural similarity is limited. Given that this is the currently most complete structural model of the myosin ATPase cycle at high resolution, we consider it a valuable contribution to the myosin field.

Yet, we see the reviewer’s points, that the paper is overlong and that the structural model might distract from the original data presented in the paper. We have therefore decided to remove the complete section, including Video 1, Figure 11 and its supplements from the manuscript. We have adjusted the abstract, introduction and summary accordingly.

Overall, this is a solid paper, with newer higher resolution structures of the rigor and ADP actomyosin V states, which will be interesting to the field, demonstrate that the actomyosin interface is essentially highly similar, and that any structural variability mostly resides in the lever.

We thank the reviewer for the appreciation of our work.

The paper does not ‘'provide unprecedented insight into force generation’' as claimed as we still do not know how myosin attaches to actin and what structural changes occur that accelerate of Pi release. This is still the key missing step. However, the new high-resolution structures of ADP and rigor myosin V bound to actin are a useful contribution to the field.

Considering that we report the first high-resolution AppNHp-myosin structure and are the first to describe the myosin-V’s structural heterogeneity at molecular detail, we believe that we have indeed provided unprecedented insights into the structural basis of force generation. We have yet down-phrased this sentence to “The presented high-resolution cryo-EM structures of the actomyosin-V complex in three nucleotide states – nucleotide-free, Mg^2+^-ADP and Mg^2+^-AppNHp – (Table 1) provide valuable insights into the structural basis of force generation.”.

There is very little experimental justification (biochemical experiments) that demonstrate the AppNHp state is indeed a post-rigor transitional state. Either more work needs to be performed to show that this is the case, or this claim needs to be scaled back.

See answer for this point provided above.

In examining heterogeneity, the data for each actomyosin state could have been divided up into more classes with a lower number in each class, which might be expected to show an increase in heterogeneity. How was the decision made on the number of classes? Did the different number of particles available for the AppNHp class affect this analysis? The authors should explain their reasoning here in the revised paper.

Choosing the number of classes was not straightforward due to the nature of the conformational heterogeneity. In the methods (also referred to in the main text, see line 494 of the original manuscript), we describe our approach and reasoning in detail. In summary, we ran multiple 3D classifications asking for varying numbers of classes e.g. 3-12. First, we tried to find a stable core of classes by comparing the structures and associated particles (see Author response image 2 for heat map for ADP data). Results showed very little overlap indicating a continuous structural heterogeneity (as opposed to discrete structural states). Continuous changes are in general difficult to analyze (software development currently ongoing in several labs). Aiming for representative results suitable for analysis of motions, we optimized the number of classes experimentally to yield the highest number of classes with a resolution and map quality sufficient for atomic modelling (≤ 3.7 Å).

In Supplementary Figure 2D and E – A comparison of maps constructed from the same particle number would be a fairer and more informative comparison. This would address the question as to whether differences in the resolution of the cryoEM maps from different states are due to heterogeneity or differences in particle number.

We assume the reviewer refers to Figure 1 —figure supplement 2 D-E here. This figure is meant to show the overall resolution of structures, in particular how the resolution varies within the actomyosin complex (central part is better resolved than the periphery). Its purpose is not to compare the resolution of different data sets, as implicated by the reviewer. We also do not claim that heterogeneity causes differences in resolution. In contrast, differences in the overall, global resolution are minor and can easily be attributed to the differences in the particle number. We therefore do not think that an additional figure comparing structures with the same number of particles would be useful.

The statement that Pi release requires completion of the power stroke is incorrect, as the lead head of myo5, when bound to actin, can release Pi, but the converter still in a pre-power state conformation (Walker et al., 2000, Nature). This is an important point, and this statement needs correcting.

We assume that the reviewer refers to the statement, that the phosphate might stay bound to a second binding site until the powerstroke has completed (lines 617-620)? As the whole section including this sentence was removed, it does not require corrections anymore. We have additionally removed the reference to Reubold et al., 2003 in the introduction, as this paper describes rearrangements that would propose that powerstroke conformational changes can occur prior to Pi release. We have instead added references to Walker et al., (as suggested by the reviewer) and Rosenfeld and Sweeney (JBC, 2004), to state that Pi release occurs without completion of the powerstroke.

Regarding the statement 'Our data indicate that the powerstroke of myosin-V amounts to ~ 77°, in contrast to a previous estimate of only 58° based on medium resolution cryo-EM data (Wulf et al., 2016). It is unclear why this angle changes when everything else seems the same as before. Is it possible that this is due to a change in the way the molecules have been super-posed before looking at global movements? Could the authors clarify this and re-analyse if appropriate.

Differences are likely due to the way the PPS and strong-ADP structures were superimposed (i.e. we superimposed on the L50 domain, while Wulf et al., superimposed on all parts of myosin that interact with F-actin) and how the rotation was measured (i.e. we calculated the angle based on axes created for each helix, see Methods). As we have removed the complete motor cycle section from the revised manuscript (see response above), the comparison of angles is no longer included and does not require any adjustments/additional explanations.

Recommendations for improving the writing and presentation.The paper is overlong, and there is considerable speculation in places that is not required and which detracts from the paper.

We have significantly shortened the manuscript (from previously 23 to 19 pages) and removed speculative paragraphs as suggested by the reviewer.

– Removed section ‘Structural model of the myosin-V motor cycle’ and corresponding figures and methods

– Adjusted Abstract, Introduction, Summary and Methods accordingly

– Removed comment on line 172-174 comparing the cryo-EM strong-ADP state with the weak-ADP rigor-like crystal structure

– Removed speculation about the role of myosin V on the remodeling of the actin cytoskeleton (lines 450-456)

– Shortened comparison of the rigor cryo-EM and rigor-like crystal structure

– Shortened the description of myosin-V’s structural transition upon ADP release

– Shortened the comparison of myosin-V,IB and VI regarding their structural transition upon ADP-release, including our proposal about the relation of the extend of the lever swing and myosin’s force sensitivity

– Shortened the description of myosin’s structural heterogeneity

– Shortened summary

The authors should revise their discussion of the comparison of the Mg.ADP myosin V structures as indicated in the public review.

See answer for this point provided above.

The authors should comment on the data from Howard White, that suggests we would not expect to find a change in actin structure between ADP and rigor states, as indicated in the public review.

See answer for this point provided above.

Generally, the writing was clear, but the paper is overly long, and contains quite a bit of speculation that could easily be removed without detracting from the key findings. The authors should shorten and focus the paper.

See answer for this point provided above.

There are a few odd uses of English in places, which would be improved by careful proof reading.

As suggested, we have carefully proof read the paper.

I recommend removing Figure 11 and associated figures completely, as they do not add to the paper. In fact, I think they somewhat detract from the key findings of the paper concerning the two main ADP and rigor actomyosin states. Moreover, it is unclear how relevant the intermediate states from myosin VI are? They might be on pathway, but the sequence is not strongly conserved, so probably not. Suppl 1 and 2 (for Figure 11) – are mainly just re-summarising figures and are not needed.

Based on the reviewer’s suggestions we have removed the complete section ‘Structural model of the myosin-V motor cycle’, including Figure 11, its supplementary figures and Video 1 from the manuscript (also see answer above).

A clearer explanation of the methods for generating the PC plots would be helpful. How was the Bio3D library used?

We added additional details on how the Bio3D library was used to the methods section (see below). In general, we followed the workflow described in the official tutorial found on http://thegrantlab.org/bio3d/.

“Conformational changes and structural heterogeneity of the central 1er models were characterized by principal component analysis (PCA) using the Bio3d library (Grant et al., 2006) in R (R Core Team, 2017). Initially, model sequences were aligned using the *pdbaln* method. With the help of the methods *core.find* and *pdbfit*, models were then superimposed on an automatically determined structural stable core, which encompasses almost the complete F-actin subunit and parts of the HLH-motif in the L50 domain. PCA was performed running *px.xray,* excluding gaps within the sequence and ligands. Data points were manually grouped and colored based on the underlying data set and type of model, i.e. average model vs. 3D class average. For the direct visualization of PCA results, trajectories along each principle component were exported using *mktrj.pca* and morphed in UCSF Chimera (Pettersen et al., 2004).”

Reviewer #2:This manuscript presents a detailed structural model of the myosin-V motor cycle by characterizing three actin-bound structural intermediates with different nucleotide states (strong-ADP, nucleotide free (rigor) and non-hydrolyzable ATP (post-rigor transition)) at near atomic resolution. These results supports and expands on details of the myosin-V mechanochemical cycle that could be inferred from previous structural work, including x-ray crystal structures and lower resolution cryo-EM studies of myosin-V. Additionally, a novel conformation is reported of ATP-analog bound, actin bound myosin-V, exhibiting striking structural similarities with the rigor state. Notably, this is the first high-resolution structure of ATP-analog bound myosin on an actin filament that has been reported. This conformation is interpreted by the authors as a post-rigor transition (PRT) state that could rationalize how ATP recognizes the rigor state before promoting detachment from F-actin. Finally, by collecting large datasets the authors were also able to generate an assessment of heterogeneity/mobility of these states which informs structural transitions between these states.This manuscript is packed with a lot of details (and structures) but organizes them nicely in the context of existing literature, avoids overinterpretation and overall represents a solid contribution to the field.

We thank the reviewer for acknowledging our work, particularly the novelty of our AppNHp-bound structure and our efforts to organize a total of 27 structures in an accessible way.

Overall the manuscript is highly polished. The readability could benefit by trimming or moving to the supplementary material several main figures that are not central to the main findings.The following points should be addressed to improve the work:1. One conclusion from the manuscript, quoted from the summary is that "Structural transitions of myosin-V are hence likely not initiated by binding of a specific nucleotide, but rather by thermodynamic fluctuations, as previously suggested for myosin-VI". This statement doesn't seem consistent with the PCA results (Figure 10 A) showing that the ADP conformational space does not strongly overlap with that of rigor. Neither does this statement appear consistent with the authors' proposal that PRT state isomerizes after nucleotide binding to release from actin.

We do not see an inconsistency here, as we solely refer to the initiation of the transition. The fact, that the conformational spaces of the strong-ADP and rigor do not overlap is not surprising, giving that we have strongly oversaturated myosin with ADP to reach full saturation (mM concentrations). While the transition to the rigor state is initiated by thermodynamic fluctuations – based on the conformational space and the nature of structural changes within-, it cannot progress further as ADP never gets released.

To clarify this point, we have added the following sentence to the manuscript

“The fact, that the conformational spaces of the strong-ADP and rigor state do not overlap is anticipated, given that we have over-saturated myosin with Mg^2+^-ADP (see Methods)”

In case of AppNHp, we believe that thermal fluctuations initiate and promote the transition to a strong nucleotide coordination, which eventually results in large structural changes and detachment from F-actin. The fact, that we find AppNHp-myosin either strongly bound to F-actin (PRT state) or completely detached, suggests that the structural transition causing detachment is too fast to capture by standard cryo-EM (but requires time-resolved experiments). To test this hypothesis and the structural state of the detached AppNHp-bound myosin, we have tried to solve the corresponding structure – unfortunately without success (see Figure 4—figure supplement 3).

Likewise, for page 17 where it is stated that "Our data show that the conformational heterogeneity of myosin-V is not caused by variations of the active site or mixed nucleotide states (Figure 9). Nevertheless, the presence of a nucleotide does affect the extent of flexibility, as ADP and AppNHp lead to a greater change in lever arm position (Figure 9A)." These statements seem contradictory and the authors need to clarify the ambiguity.

We find that the statement on page 17 is correct and not confusing. It states that the nucleotide is not the cause of the flexibility, but biases its amount; in other words, the overall extent of flexibility is larger when ADP or AppNHp is bound. The recovery-stroke is a good example of the role of thermodynamic fluctuations in structural transitions, since it occurs while ATP is bound to active-site. Based on our data, thermodynamic fluctuations are also essential for other transitions, but the binding of a specific nucleotide biases the likelihood of these transitions and their kinetics.

We understand that this point might be difficult to understand and also agree that the statement in the summary (referred to in the paragraph above) might be misleading due to its generalization and simplification. We have therefore changed it as follows

“While the extent of flexibility is altered by the presence of a nucleotide, structural transitions of myosin-V are likely not initiated by binding of a specific nucleotide, but rather by thermodynamic fluctuations, as previously suggested for myosin-VI (Blanc et al., 2018).

2. Under the "Conservation and specificity of the actomyosin-V interface" section on page 11, the last sentence says "Differences in the affinity might therefore not be linked to altered contacts, but rather to the degree of structural flexibility inherent to each state (see below)". This is an interesting statement, but the discussion below or in other sections don't address this. The authors need to expand on this.

We briefly discuss this point in the ‘Pronounced structural heterogeneity of myosin-V’ section (see below). The key point is that strong flexibility is thought to be detrimental for tight binding. The larger flexibility of the strong-ADP state could therefore account for its lower binding affinity for F-actin.

“It also provides a good explanation for the different binding affinities of the rigor and strong-ADP state. Specifically, we propose that the extent of conformational heterogeneity tunes the binding affinity, rather than changes in the actomyosin interface since these are almost the same in all three nucleotide states studied (Figure 5).”

3. The atomic model of strong ADP myosin-V is one of the more significant results since only two other myosins have been solved to near atomic resolution in this nucleotide state. It therefore seems important to provide a comparison of active site features (ie., Magnesium coordination etc) for the available ADP actomyosin co-complex structures, at least as a supplementary figure.

In the revised manuscript we include a new supplementary figure (Figure 1—figure supplement 2) showing a comparison of the Mg^2+^-ADP coordination in myosin-V, myosin-IB, myosin-VI (although the model does not contain a nucleotide) and myosin-XV (paper on BiorXiv, but model already released). We have adjusted the main text as follows

“The coordination of Mg^2+^-ADP in our structure closely resembles the ones reported for the strong-ADP state of myosin-IB (Mentes et al., 2018), myosin-VI (Gurel et al., 2017) and myosin-XV (Gong et al., 2021) (Figure 1—figure supplement 2). Solely the position of switch I differs appreciably between myosins, ultimately resulting in varying positions of the coordinated Mg^2+^ ion.”

Reviewer #3:The paper by Pospich provides important new structural insights into actomyosin-V mechanochemical coupling. The work provides new high-resolution structures of several states of myosin-V, provides new details on the relationship between myosin-binding and actin structure, and gives interesting insights into the structural heterogeneity of myosin-V.The paper is written in a way that provides testable predictions of the importance of specific residue-residue interactions to nucleotide binding, and the conformational changes that result in lever arm tilting upon ADP release. A strength of the paper is the thoughtfulness of the supplemental videos that relate to the figures.

We thank E. Michael Ostap for appreciating the quality of our work as well as our efforts to improve accessibility by carefully designed figures and videos.